# An active tethering mechanism controls the fate of vesicles

Seong J. An[1,3], Felix Rivera-Molina[1,3], Alexander Anneken[1], Zhiqun Xi[1], Brian McNellis[1], Vladimir I. Polejaev[2] & Derek Toomre[1✉]

Vesicle tethers are thought to underpin the efficiency of intracellular fusion by bridging vesicles to their target membranes. However, the interplay between tethering and fusion has remained enigmatic. Here, through optogenetic control of either a natural tether—the exocyst complex—or an artificial tether, we report that tethering regulates the mode of fusion. We find that vesicles mainly undergo kiss-and-run instead of full fusion in the absence of functional exocyst. Full fusion is rescued by optogenetically restoring exocyst function, in a manner likely dependent on the stoichiometry of tether engagement with the plasma membrane. In contrast, a passive artificial tether produces mostly kissing events, suggesting that kiss-and-run is the default mode of vesicle fusion. Optogenetic control of tethering further shows that fusion mode has physiological relevance since only full fusion could trigger lamellipodial expansion. These findings demonstrate that active coupling between tethering and fusion is critical for robust membrane merger.

[1] Department of Cell Biology, Yale University School of Medicine, New Haven, CT, USA. [2] International Science and Technology Center, Yale University School of Medicine, New Haven, CT, USA. [3] These authors contributed equally: Seong J. An, Felix Rivera-Molina. ✉email: derek.toomre@yale.edu

Vesicle tethers are protein complexes that physically connect a transport vesicle to its target membrane prior to fusion. Acting upstream of the SNARE fusion machinery, tethers are thought to mediate the initial interaction between membranes that are destined to merge. The importance of tethering is highlighted by the fact that nearly every intracellular trafficking pathway, from yeast to human, critically depends on a dedicated vesicle tether[1]. Yet, because it is well known that membrane fusion in vitro does not strictly require vesicle tethers[2], the importance of tethering in cells remains unclear. The conventional view is that vesicle fusion in cells is much more complicated than it is in vitro and requires tethers to impart an efficiency or fidelity to the process that may not be obvious[3]. For example, it has been suggested that tethers may play a role in the kinetic regulation[4–8] or quality control[9–12] of membrane transport events. Such studies have recognized, explicitly or not, that the role of tethers is likely more than simply bridging membranes since this task is naturally performed by the fusion machinery itself to execute membrane fusion[1]. Indeed, the incidental tethering caused by the fusion machinery could explain why bona fide tethering is not absolutely necessary in reconstituted fusion assays.

In cells, however, tethers connect vesicles to specific places such as organelles. Consideration of a spatial aspect to tethering allows for two fundamentally different models of how tethers could promote membrane fusion. In a passive mechanism, assuming that fusion sites in cells are privileged[13], tethers could merely hold vesicles near their destination to increase the probability of fusion[3]. This scenario also assumes that fusion is rate-limited by the availability of fusion machinery components at the fusion site. Alternatively, in an active mechanism, tethers could promote fusion by directly or indirectly engaging with the fusion machinery to regulate its formation. Passive and active tethering can be considered as purely kinetic and thermodynamic processes, respectively[3]. Evidence for both models has been reported[5,7,8,14–17], and it may be that not all tethering events use the same mechanism to promote fusion. Nonetheless, given that all intracellular fusion events use a universal mechanism[1], it is reasonable to think that general principles of tethering should also exist.

Originally discovered over two decades ago[18], the exocyst is perhaps the best characterized vesicle tether. It is an evolutionarily conserved, hetero-octameric complex that plays an essential role in exocytosis[19]. It is required in trafficking pathways that support diverse cellular processes such as cell migration[20–23], glucose transport[24], cytokinesis[25], ciliogenesis[26–28], autophagy[29] and cell survival[30]. In the "spatial landmark" model[31], the exocyst is thought to initiate tethering when its Exo70 and Sec3 subunits, located on the plasma membrane by virtue of their ability to bind lipids, assemble into a holocomplex with the remaining six subunits (Sec5, Sec6, Sec8, Sec10, Sec15 and Exo84) that arrive on vesicles through attachment to the small GTPase Sec4p in yeast[32] and Rab11 in mammals[33]. The exocyst may then promote fusion, at some point during or after its assembly, by interacting with components of the fusion machinery[14,34]. Notably, while the core mechanics of this model—that physical tethering by the exocyst leads to fusion—is 15 years old, it still remains unverified by direct experimental evidence. A major problem is that conventional methods of studying tethering use vesicle fusion as a functional readout, which only provides indirect information about the tethering process. Furthermore, the function of tethers is largely studied by chronically silencing their components or expressing their mutants, which may produce long-term, non-physiological effects on tethering. We[35] and others[36] have monitored tethering in live cells by imaging exocyst subunits relative to vesicle fusion, but such spatiotemporal correlations alone do not elucidate the causal relationship between tethering and fusion. For these reasons, it would be helpful not only to monitor tethering but also to control tethering in real time.

Exocytosis is initiated when a vesicle forms a fusion pore with the plasma membrane. Once formed, the fusion pore can either rapidly dilate or reseal, resulting in the vesicle fully integrating with the plasma membrane or retaining its gross morphological shape through processes termed full fusion (FF) and kiss-and-run (KR), respectively. The mode of vesicle fusion may impact key aspects of exocytosis such as vesicle recycling and release dynamics of vesicle cargo[37]. While it is an unconventional type of exocytosis whose mechanism is unclear, KR has been documented in a wide variety of cell types, including neurons[38,39] and non-neuronal cells[40]. Whether a relationship exists between fusion mode and vesicle tethering is currently unappreciated.

Here we combine live-cell imaging and optogenetic control of tethers to study the role of the exocyst in tethering during exocytosis. We unexpectedly discover that FF is intrinsically inefficient: without the exocyst or when Exo70 is mutated, vesicles mainly undergo KR. By optogenetically controlling tethering with two different light-dependent heterodimerization systems[41,42], we show that exocyst-mediated tethering acutely rescues FF, in a manner that depends on the number of exocyst complexes per vesicle that engage with the plasma membrane. In contrast, passively tethering vesicles with a simple, artificial tether fails to promote FF. We further show that the mode of fusion during exocytosis has physiological consequence as FF but not KR promotes membrane remodeling. Collectively, our findings demonstrate that an active tethering mechanism controls the mode of vesicle fusion, thus revealing the functional importance of vesicle tethering in cells.

## Results

**Membrane binding by Exo70 influences tethering duration**. The exocyst is required for exocytosis of post-Golgi vesicles in yeast[19] and endocytic recycling vesicles in mammalian cells[33]. In agreement, we previously showed that when Sec8-RFP molecularly replaces endogenous Sec8 in HeLa cells, it is mainly found on recycling vesicles, based on its high colocalization with the transferrin receptor and Rab11, but low colocalization with the post-Golgi markers VSVG and NPY[35].

Because Exo70 was proposed to serve as a spatial landmark for vesicle tethering at the plasma membrane of yeast[31,43] and mammalian[44] cells, we examined its localization in HeLa cells. Using widefield deconvolution microscopy, we found that endogenous Exo70 largely colocalized with Sec8-RFP on Rab11-positive vesicles (Fig. 1a, solid arrowheads). Knockdown (KD) of Sec15, the subunit that mediates exocyst–Rab11 interaction[45], reduced the pairwise colocalization of both Sec8-RFP and Exo70 with Rab11 (Fig. 1a, Pearson's correlation graphs). Because Exo70 and Sec8 reside on different halves or subcomplexes of the exocyst[46,47], this suggests that the exocyst is fully assembled on vesicles before tethering even occurs, contrary to the spatial landmark model. Indeed, the presence of the exocyst holocomplex on vesicles, for which evidence already exists[36,48], may explain why most subunits mislocalize to the cytosol when overexpressed[49], but can properly localize on vesicles if the corresponding endogenous subunit is silenced[35].

Although endogenous Exo70 did not mark the plasma membrane as expected from the spatial landmark model[31], we explored the long-held idea that membrane binding by Exo70 serves to tether vesicles physically[31,43,44]. Exo70 possesses four domains that form an extended rod[50]. Its C-terminal domain binds phosphatidylinositol 4,5-bisphosphate [PI(4,5)P$_2$] through a patch of basic residues[43,44]. Inhibiting this binding by mutating

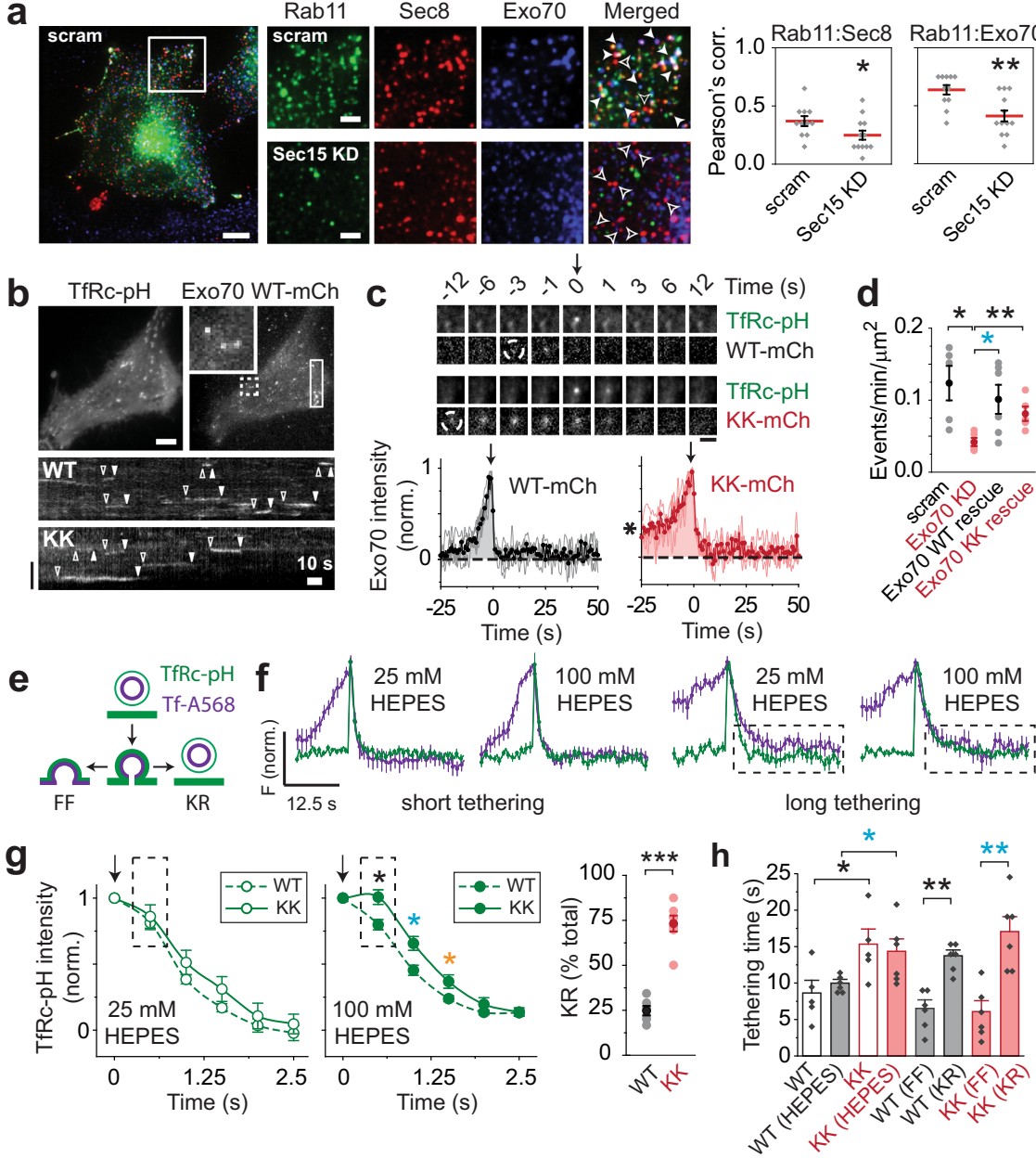

**Fig. 1 Membrane binding by Exo70 influences the mode of vesicle fusion. a** Localization of endogenous Exo70 in HeLa cells. Solid and open arrowheads, selected examples of triple-colocalizing and Sec8-only spots, respectively (left). Scale bars, 6 and 2 μm for whole and zoomed images, respectively. Pearson's correlation of Exo70 and Sec8 with Rab11 (right). Data represent mean ± SEM (*$P = 0.049$, **$P = 0.0017$, two-tailed Student's *t*-test). **b** Two-color TIRFM of Exo70-WT–mCherry and transferrin receptor-pHluorin (TfRc-pH). Maximum-intensity projection of movie (top). Scale bar, 6 μm. Inset, zoom of dashed box. Kymograph for Exo70-WT (middle; region outlined by rectangle) and -KK (bottom). Arrowheads, tethering duration. Vertical scale bar, 6 μm. **c** Average image sequence of fusing vesicles from one cell (top). Arrow, fusion onset. Dashed white circles, initial tethering. Scale bar, 2 μm. Exo70-WT (black) and Exo70-KK (red) traces, time aligned to fusion (bottom). Averages (bold line) of cell averages (light lines) are shown. $n = 5$ cells for both Exo70-WT and -KK. Asterisk denotes elevated tethering signal for Exo70-KK. **d** Rescue of vesicle fusion by Exo70 constructs. $n = 5$ cells for scram, Exo70 KD, Exo70-KK rescue, and $n = 6$ cells for Exo70-WT rescue. Mean ± SEM (*$P = 0.0011$, cyan *$P = 0.029$, **$P = 0.0091$, two-tailed Student's *t*-test). **e** Schematic of TfRc-pH (green) and transferrin (Tf)-A568 (purple) signals during full fusion (FF) and kiss-and-run (KR). **f** Effect of 100 mM HEPES on vesicle reacidification (compare dashed boxes). $n = 26$ vesicles for short tethering with both 25 and 100 mM HEPES, and $n = 24$ and 40 vesicles for long tethering with 25 and 100 mM HEPES, respectively. Mean ± SEM. **g** Effect of 100 mM HEPES on the average TfRc decay with Exo70-WT and -KK (left). $n = 5$ cells for both Exo70-WT and -KK in 25 mM HEPES, and $n = 6$ cells for both Exo70-WT and -KK in 100 mM HEPES. Mean ± SEM (*$P = 0.011$, cyan *$P = 0.011$, orange *$P = 0.42$, two-tailed Student's *t*-test). Arrow, fusion onset. Dashed box highlights difference in decay for the first time point (0.5 s) after fusion onset. Frequency of KR with Exo70-WT and -KK (right). Mean ± SEM (***$P = 2.2 \times 10^{-6}$, two-tailed Student's *t*-test). **h** Summary of tethering times. Mean ± SEM (*$P = 0.038$, cyan *$P = 0.048$, **$P = 0.0011$, cyan **$P = 0.0015$, two-tailed Student's *t*-test). Number and sample size of experiments, here and elsewhere, are shown in Supplementary Table 1.

two conserved lysines to alanines (K632A, K635A) blocks the surface delivery of secretory cargo, presumably by preventing vesicle tethering[44]. We confirmed that wildtype Exo70 (Exo70-WT), but not a variant carrying these mutations (Exo70-KK), could bind to the plasma membrane when overexpressed (Supplementary Fig. 1). Both Exo70-WT and -KK could be incorporated into the exocyst complex by molecular replacement of endogenous Exo70 (i.e., during Exo70 KD; Supplementary Fig. 2).

Live-cell imaging by total internal reflection fluorescence (TIRF) microscopy revealed that both Exo70-WT and -KK in Exo70 KD cells decorated spots that appear briefly at the plasma membrane (Fig. 1b), similar to previously observed dynamics of Sec8 on vesicles[35]. By coexpressing transferrin receptor-pHluorin (TfRc-pH), a pH-sensitive exocytosis reporter[51], we could ascertain whether visitations of Exo70 spots at the membrane reflected the tethering of vesicles before fusion. To our surprise, we found this to be the case not just for Exo70-WT but also for Exo70-KK (Fig. 1c, Supplementary Fig. 3, and Supplementary Movie 1). This suggests that exocyst complexes reconstituted with Exo70-KK retain their ability to tether vesicles, which was further supported by the partial rescue of the fusion rate in Exo70 KD cells by Exo70-KK (Fig. 1d).

For Exo70-labeled vesicles, individual tethering events resembled step functions when plotted against time (Supplementary Fig. 3b), but they produced a smoothly rising trace when averaged (after temporal alignment to fusion onset) as individual tethering times varied. How quickly an averaged trace rose thus is a measure of tethering duration, which was noticeably longer for Exo70-KK than it was for either Exo70-WT (Fig. 1c), Sec8 or Rab11 (Supplementary Fig. 4). At 25 s before fusion, for instance, the Exo70-WT trace was near zero baseline, but the Exo70-KK trace was already at ~25% of peak intensity (Fig. 1c), indicating that a greater fraction of vesicles was tethered for more than 25 s with the Exo70 mutant.

To monitor tethering without expressing fluorescently tagged exocyst subunits or Rab11, which might inadvertently affect the tethering process, we imaged TfRc conjugated to pHTomato[52], a pH-sensitive reporter that is incompletely quenched within vesicles and thus allows their detection prior to fusion. With TfRc-pHTomato, the tethering duration was similar to those observed with tagged Exo70-WT, Sec8 and Rab11. However, when Exo70 was knocked down, the duration became anomalously long (Supplementary Fig. 5). Evidently, the exocyst confers short tethering times when it contains Exo70-WT but not Exo70-KK. Together, these results suggest that membrane binding by Exo70 is not necessary for vesicle tethering, but rather surprisingly influences the duration of vesicle tethering.

**Membrane binding by Exo70 affects the mode of fusion.** Despite their markedly different average traces, both Exo70-WT and -KK mediated short and long tethering events, but to differing degrees. Interestingly, irrespective of the Exo70 variant, long tethering was correlated with a slower, non-spreading decrease of TfRc-pH fluorescence after fusion (Supplementary Fig. 3). This suggests that long tethering events might have culminated in KR rather than FF. To investigate this possibility rigorously, we tested the sensitivity of TfRc-pH, after it brightens during fusion, to high extracellular HEPES (100 mM), which can diffuse through the fusion pore and retard vesicle reacidification if the vesicle undergoes KR[38,53] (Fig. 1e). As an internal control, we additionally labeled vesicles with transferrin (Tf) ligand conjugated to a pH-insensitive dye (Tf-Alexa568). Using this approach, we discerned two classes of fusion events: (i) fast decay of receptor and ligand fluorescence after short tethering and (ii)

slow decay of both after long tethering (Fig. 1f and Supplementary Fig. 6). Importantly, 100 mM HEPES affected TfRc-pH fluorescence only with Class ii events, ablating the faster decay of the receptor compared to ligand in 25 mM HEPES (dashed boxes in Fig. 1f). This demonstrates that the non-spreading decrease of TfRc-pH fluorescence associated with long tethering reflects both vesicle reacidification and the departure, or run, of a vesicle from the membrane after it transiently fuses, or kisses.

To distinguish fusion modes using another approach, we took line-intensity profiles of Tf ligand before and during fusion and fitted these profiles with Gaussian functions. Consistent with their differential sensitivity to high HEPES, Class i but not Class ii events showed a progressive increase in the width of the Gaussians curves after fusion (Supplementary Fig. 7), which is indicative of FF[54]. Thus, two independent approaches of discerning fusion modes—sensitivity of TfRc-pH to extracellular HEPES buffer and lateral spread of vesicle cargo (Tf ligand)—show that short tethering times are associated with FF, while long tethering times are associated with KR.

Having ascertained that recycling vesicles can undergo KR, we next examined the fusion modes mediated by Exo70-WT and -KK. As expected, high HEPES increased the difference in the average decay of TfRc-pH between Exo70-WT and -KK fusion events (dashed boxes in Fig. 1g, left). The slower decay with Exo70-KK reflected a higher frequency of KR (Fig. 1g, right), which was easily distinguishable from FF in 100 mM HEPES (Supplementary Fig. 8). Quantification of the overall and class-specific tethering times with Exo70-WT and -KK (Fig. 1h) recapitulated the correlation between tethering duration and fusion mode seen with Tf receptor and ligand: it takes longer for vesicles to kiss than to fuse fully. Therefore, we conclude that membrane binding by Exo70 affects not only how long vesicles are tethered but also how completely they undergo fusion.

**Exo70 optogenetics rescues full fusion.** The data presented so far suggest a role for Exo70 membrane interaction in regulating the mode of fusion. To determine whether this role is direct, we turned to an optogenetics approach. The CRY2-CIB hetero-dimerization system allows controlled juxtaposition of two components through the blue-light triggered interaction between cryptochrome 2 (CRY2), the light sensor, and the transcription factor CIB[41,55]. We reasoned that Exo70-KK–CRY2 (the optogenetic prey) binding to membrane-targeted CIB (the optogenetic bait) could rescue the ability of Exo70-KK–positive vesicles to fuse fully (Fig. 2a). It should be noted that with this strategy, 488-nm light is used both to image TfRc-pH and to activate CRY2. Unexpectedly, we found that some vesicles became stuck during an optogenetics experiment, staying at the membrane for the entire duration (~4 min) of a movie (Fig. 2b, red arrowheads; and Supplementary Movie 2). Without CIB (Fig. 2c), the appearance of vesicles at the membrane was similar to that seen in non-optogenetics experiments (Fig. 1b).

On average, optogenetically stimulated vesicles were tethered for a long time not only before they fused but also ostensibly afterwards (Fig. 2c). To confirm that the post-fusion Exo70-KK–CRY2 signal represented vesicles, rather than merely Exo70-KK–CRY2 alone (e.g., as an aggregate that remained near the fusion site), we again employed 100 mM HEPES to impede vesicle reacidification. Strikingly, the buffer permanently elevated the average TfRc-pH signal after fusion above zero baseline (Supplementary Fig. 9). Closer inspection of individual fusion events indicated that this effect reflected the presence of a third type of fusion mode, one that is specific to optogenetically induced tethering, which we term kiss-and-stay (KS; Fig. 2d, e). Stuck vesicles thus represented the inability of vesicles undergoing

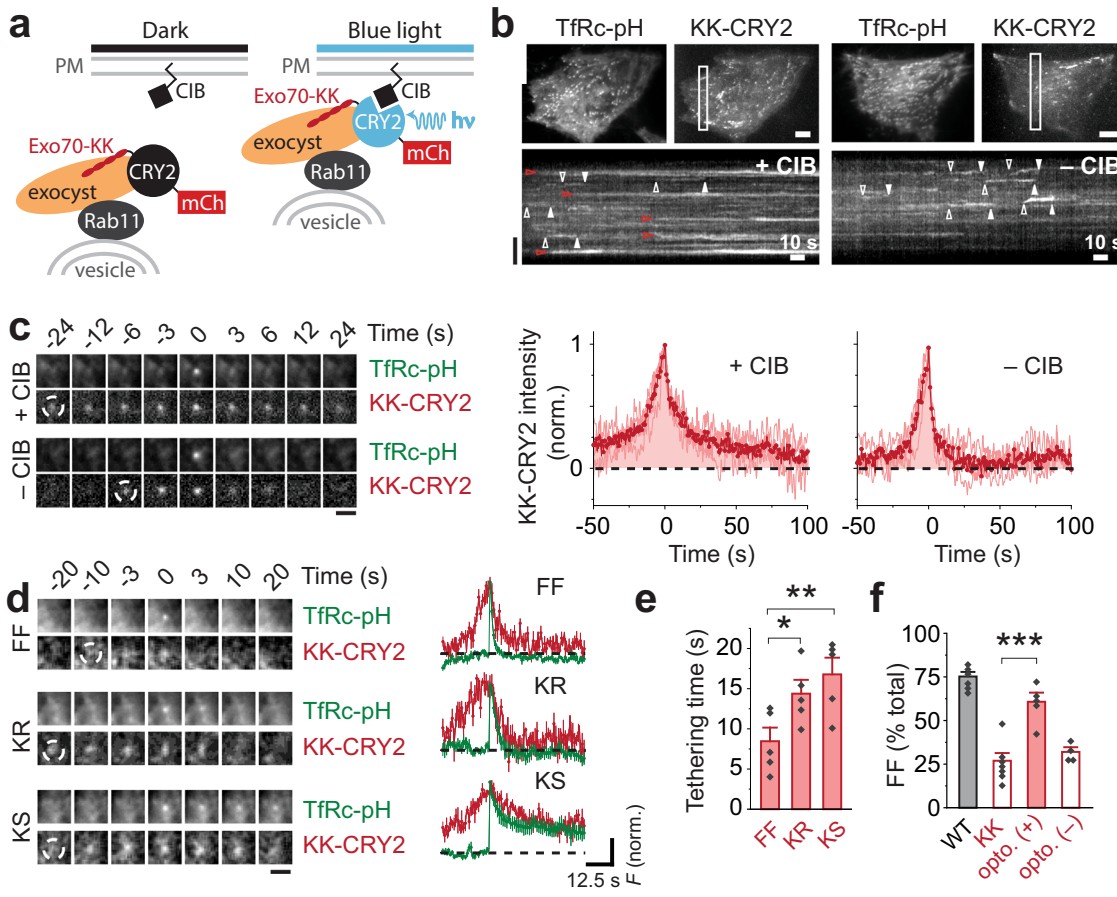

**Fig. 2 Exo70 optogenetics rescues the fusion mode defect of Exo70-KK. a** Schematic of Exo70 optogenetics using the CRY2-CIB system. PM, plasma membrane. **b** Exo70-KK–CRY2 optogenetics with or without CIB. Cells were activated at 2 Hz with 100-ms pulses of 488-nm light (1.5 W/cm$^2$). Maximum-intensity projection of movies (top). Scale bars, 6 µm. Kymographs of Exo70 channel (bottom). Red arrowheads, stuck vesicles. Vertical scale bar, 6 µm. **c** Average image sequence of fusing vesicles from one cell (left). Scale bar, 2 µm. Exo70-KK–CRY2 traces (red), time aligned to fusion (right). Averages (bold line) of cell averages (light lines) are shown. $n = 5$ and 4 cells for + CIB and – CIB, respectively. **d** Three fusion modes (FF, KR and KS) observed with Exo70-KK–CRY2 activation (+CIB) using 100 mM HEPES. Average image sequence of FF, KR and KS events from one cell (left). Scale bar, 2 µm. TfRc-pH (green) and Exo70-KK–CRY2 (red) traces (average of cell averages) for fusion modes (right). Mean ± SEM. Black dashed lines, zero baseline. **e** Tethering half-times for different fusion modes ($n = 6$ cells). Mean ± SEM (*$P = 0.013$, **$P = 0.0035$, two-tailed Student's $t$-test). **f** Rescue of FF by Exo70 optogenetics with (+) CIB, but not without (–) CIB. $n = 6$ cells for Exo70 WT, $n = 7$ cells for Exo70 KK, $n = 5$ cells for Exo70 optogenetics + CIB and $n = 4$ cells for Exo70 optogenetics – CIB. Mean ± SEM (***$P = 5.9 \times 10^{-4}$, two-tailed Student's $t$-test).

KR to run away due to persistence of the CRY2-CIB interaction. Nevertheless, the majority of vesicles could be induced to undergo FF (60.8 ± 5.2%; Fig. 2f), demonstrating that membrane binding by Exo70-KK–CRY2 had overcome the functional deficiency of Exo70-KK.

**Exo70 optogenetics using an independent hetero-dimerization system.** A potential complexity of the CRY2-CIB dimerization system is that CRY2 itself can homo-oligomerize upon light activation to form clusters[56], particularly when it is localized on membranes[57]. Therefore, it is possible that clustering of Exo70-KK–CRY2, were it to occur, might affect the tethering process and thus somehow influence our optogenetics results. To rule out this possibility, we repeated the experiments using a different optogenetics system. In the iLID system[42], hetero-dimerization occurs between (i) the bacterial peptide SsrA, which is embedded in the C-terminal helix of the photoreceptor – the light-oxygen-voltage2 (LOV2) domain from *Avena sativa*—and (ii) its inter-acting partner SspB. In the dark, the SsrA peptide is sterically prevented from binding SspB; however, when activated by blue light, LOV2 releases its C-terminal helix, which allows the SsrA

peptide to bind SspB. Importantly, LOV2 does not homo-oligomerize in the dark or when activated by light[42].

To optogenetically tether vesicles with the iLID system, we expressed Exo70-KK–mCherry-SspB and membrane-targeted LOV2-SsrA (designated "iLID" in Fig. 3a). Similar to the Exo70-KK–CRY2 system, activation with light caused vesicles to become stuck but, as expected, only when the LOV2-SsrA bait was coexpressed (Fig. 3b, c). These stuck vesicles once again reflected KS events (Fig. 3d), as the fluorescence of TfRc-pH remained elevated in 100 mM HEPES after fusion, compared to FF or KR events (Fig. 3e). Importantly, Exo70-KK–mCherry-SspB rescued FF, which occurred with a frequency of 64% compared to 33% ($P = 7 \times 10^{-4}$) without coexpression of LOV2-SsrA (Fig. 3f, right). Furthermore, the average tethering time for FF was again shorter than that of either KR or KS events (Fig. 3f, left), demonstrating that the correlation between tethering time and fusion mode is robust. Thus, Exo70-KK–mCherry-SspB recapi-tulated the main findings obtained with Exo70-KK–CRY2 (Fig. 2) —namely, the rescue of FF and the production of a minor fraction of KS events. This suggests that the results obtained with Exo70-KK–CRY2 mainly reflect its binding to CIB during tethering and

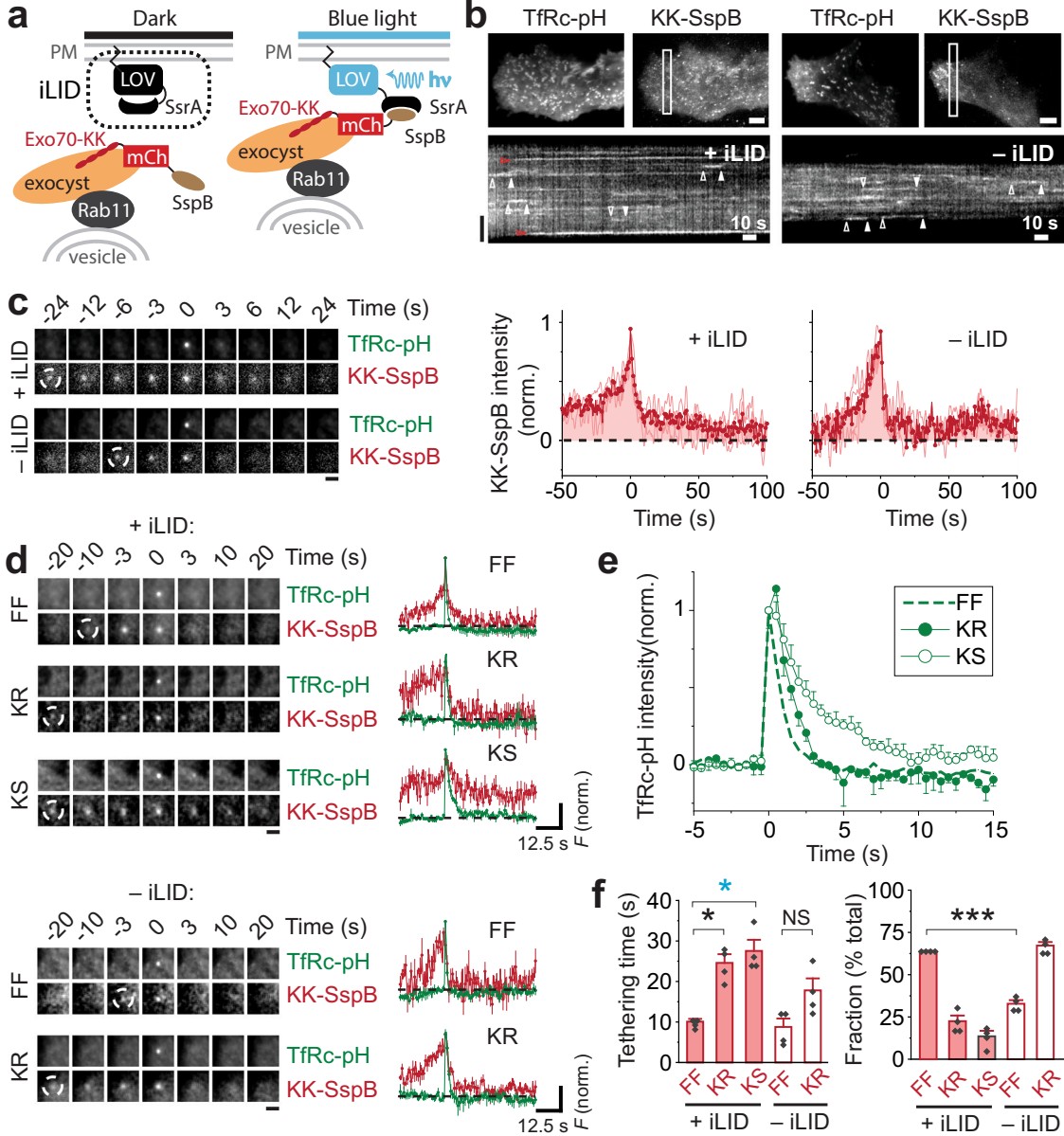

**Fig. 3 Exo70 optogenetics with the iLID system recapitulates tethering with the CRY2-CIB system. a** Schematic of Exo70 optogenetics using the iLID system. **b** Exo70-KK–mCherry-SspB optogenetics with or without LOV2-SsrA ("iLID"). Cells were activated at 2 Hz with 100-ms pulses of 488-nm light (1.5 W/cm$^2$). Maximum-intensity projection of movies (top). Scale bars, 6 µm. Kymographs of Exo70 channel (bottom). Red arrowheads, stuck vesicles. Vertical scale bar, 6 µm. **c** Ensemble average image sequence of fusing vesicles from four cells (left). Scale bar, 2 µm. Exo70-KK–mCherry-SspB traces (red), time aligned to fusion (right). Averages (bold line) of cell averages (light lines) are shown. $n = 4$ cells for both + iLID and – iLID optogenetics. **d** Three fusion modes (FF, KR and KS) observed with coexpression of iLID (top) and two fusion modes (FF and KR) observed without coexpression iLID (bottom) using 100 mM HEPES. Ensemble average image sequences from four cells for both iLID conditions (left). Scale bars, 2 µm. TfRc-pH (green) and Exo70-KK–mCherry-SspB (red) traces (average of cell averages) for different fusion modes (right). Mean ± SEM. Black dashed lines, zero baseline. **e** TfRc-pH traces (green) from the + iLID condition in **d** overlaid on an expanded timescale. Note that the TfRc-pH signal for KS remains elevated after fusion. **f** Tethering half-times for different fusion modes (left). Mean ± SEM (*$P = 0.015$, cyan *$P = 0.012$, NS = not significant, two-tailed Student's t-test). Rescue of FF by Exo70 iLID optogenetics (right). Mean ± SEM (***$P = 7.5 \times 10^{-4}$, two-tailed Student's t-test).

not any potential homo-oligomerization. However, if clustering of CRY2 does occur, it apparently has no appreciable effect on tethering.

**Full fusion depends on the degree of Exo70 membrane engagement.** Why does KS occur in Exo70 optogenetics experiments? The simplest possibility is that tethering and FF require a different number of tethers to engage with the membrane. In

other words, KS may result from suboptimal activation of tethering. To test this, we explored the dose-response relationship between light intensity and fusion mode in the Exo70-KK–CRY2 system. Remarkably, we found that even though light activated vesicles in a saturable manner (Fig. 4a, thick dashed line), it continuously modulated the mode of fusion, selectively inducing KS at the lowest intensity and FF at the highest intensity. This suggests that KS is optimally promoted by an

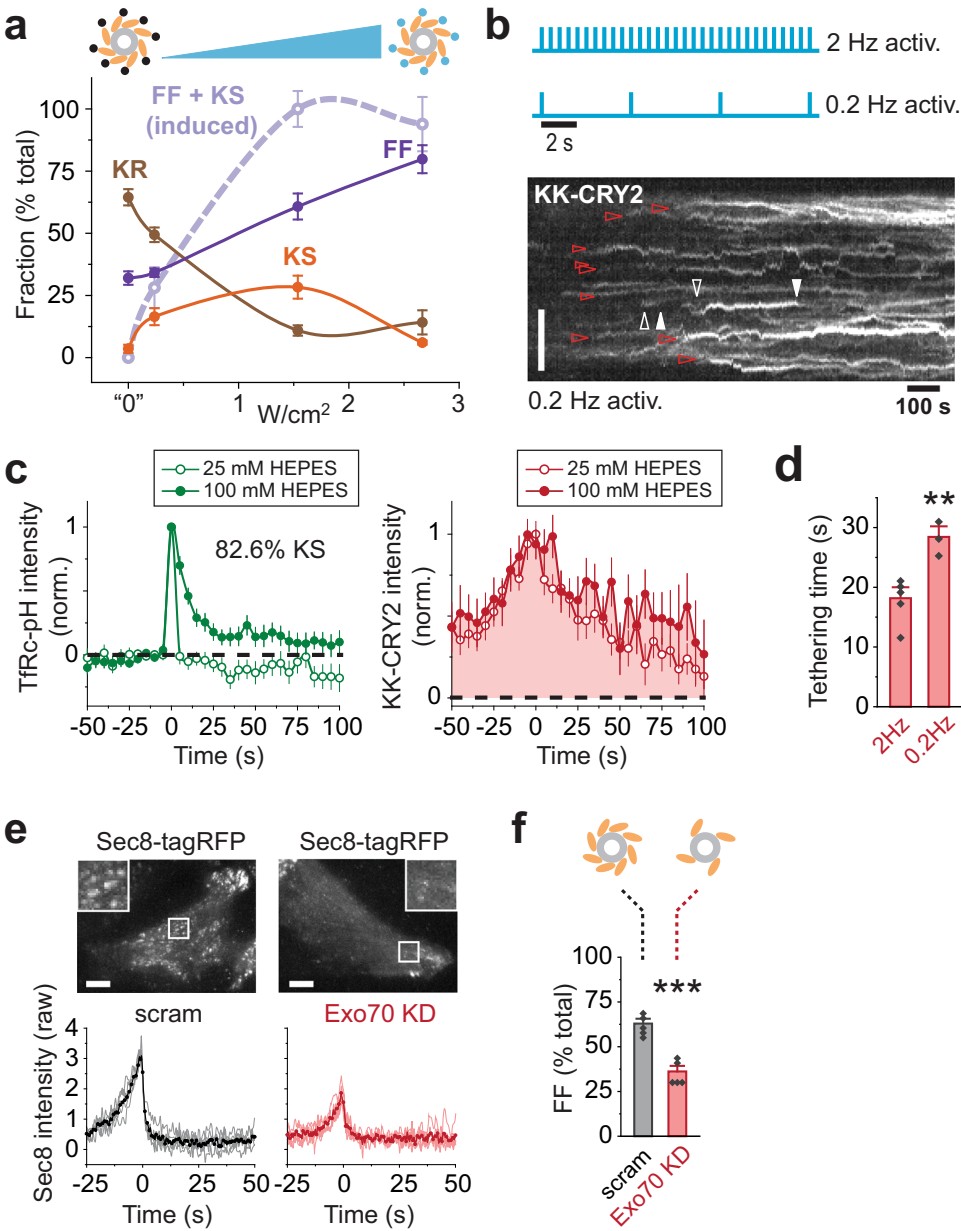

**Fig. 4 Mode of vesicle fusion depends on degree of Exo70 membrane engagement. a** Dose–response relationship between light intensity and fusion mode with Exo70-KK–CRY2. Laser power densities used were 0.23, 1.5 and 2.6 W/cm². "0" intensity activation reflects the –CIB condition. $n = 4$ cells for 0 intensity and $n = 5$ cells for all other intensities. Cartoon depicting activation of CRY2 (black/cyan circles) attached to exocyst complexes (orange ovals) serves as a simplified interpretation. **b** Schematic of high (2 Hz) and low (0.2 Hz) activation frequencies (top). Kymograph of Exo70-KK–CRY2 during low frequency activation (bottom). Red arrowheads mark stuck vesicles. Vertical scale bar, 6 μm. **c** TfRc-pH (green) and Exo70-KK–CRY2 (red) traces produced with low frequency activation, time aligned to fusion. Averages of pooled events are shown ($n = 52$ and 36 events for 25 and 100 mM HEPES, respectively). Mean ± SEM. **d** Comparison of tethering half-times produced with high (2 Hz) and low (0.2 Hz) frequency activation of Exo70-KK–CRY2. $n = 5$ and 3 cells for 2 and 0.2 Hz activation, respectively. Mean ± SEM (***$P = 0.0010$, two-tailed Student's $t$-test). **e** Effect of Exo70 knockdown (KD) on exocyst stability. Visualization of exocyst complexes with Sec8-tagRFP (top). Scale bars, 6 μm. Sec8 traces (averages of cell averages) time-aligned to fusion with (red) or without (black) Exo70 KD (bottom). $n = 5$ cells for both scram and Exo70 KD. **f** Effect of Exo70 KD on FF frequency. Mean ± SEM (***$P = 1.9 \times 10^{-4}$, two-tailed Student's $t$-test). Cartoon depicting number of exocyst complexes serves as a simplified interpretation.

intermediate level of activation that cannot be increased without promoting FF instead. As such, we wondered whether modulating the activation frequency rather than light intensity could be a more robust method of triggering KS. Indeed, reducing the activation frequency by a factor of 10 (0.2 versus 2 Hz) produced a striking preponderance of stuck vesicles and an accordingly high KS frequency (~83%; Fig. 4b–d, and Supplementary Fig. 10). Thus, modulating either light intensity or frequency of stimulation can control the mode of fusion. We note that a lower

acquisition rate could possibly favor the detection of kissing over FF events because of the slower decay of TfRc-pH during transient fusion. However, we consider this unlikely since the tethering profiles with 25 and 100 mM HEPES were not significantly different at 0.2 Hz (Fig. 4c, left panel), despite a much slower TfRc-pH decay with the higher buffer concentration (Fig. 4c, right panel).

The above optogenetics experiments suggest that the level of Exo70 membrane engagement affects the mode of fusion. To

corroborate this notion using an independent approach, we exploited the destabilization of the exocyst complex by Exo70 KD (Supplementary Fig. 2), which was apparently incomplete in some cells. For example, Fig. 4e shows that Exo70 KD greatly reduced the appearance of Sec8 spots in double-stable Sec8-tagRFP/Sec8 KD cells. This likely reflects a reduced number of exocyst complexes on vesicles because, as previously mentioned, vesicular localization of exogenously expressed Sec8 requires molecular replacement of the endogenous subunit[35]. Nonetheless, we could clearly detect a reduced Sec8 signal on individual fusing vesicles, which was ~59% of the control signal on average (Fig. 4e, bottom). This experimental setup thus allowed us to test directly whether a smaller vesicular complement of exocyst complexes influences the mode of fusion. As expected, by using 100 mM HEPES, we found that a reduced amount of Sec8 on vesicles is associated with a decreased frequency of FF (Fig. 4f). Collectively, these data argue that the ability of the exocyst to regulate the fusion mode depends on the number of exocyst complexes that engage with the membrane via the Exo70 subunit.

**Rab11 optogenetics does not rescue full fusion**. Although Exo70 optogenetics can rescue FF, it may do so not because it restores the function of exocyst complexes containing Exo70-KK, but rather simply because it tethers vesicles incidentally through optogenetic heterodimerization of the bait and prey. If this were true, then artificially tethering vesicles in the absence of the exocyst should promote FF. To test this possibility, we depleted exocyst complexes on vesicles by Exo70 KD (as we did for the experiments described in Fig. 4e, f) and fused CRY2 directly to Rab11 (Fig. 5a), the GTPase which normally mediates the attachment of the exocyst to vesicles[33,45].

Strikingly, CRY2-Rab11 mainly produced stuck vesicles in Exo70 KD but not control (scram) cells (Fig. 5b, red arrowheads). Figure 5c shows an example of a very long tethering event mediated by CRY2-Rab11 in an Exo70 KD cell. Here, a vesicle is tethered close to one location for several minutes. However, lateral drifting of vesicles during tethering was common and accounted for the non-dot-like appearance of some CRY2-Rab11-labeled vesicles in the maximum-intensity projection of CRY2-Rab11 movies (Fig. 5b, upper right panel). It should be noted that vesicle drift can cause the tethering duration to be underestimated, especially for long tethering events, as the region of interest used to measure vesicle fluorescence is centered where fusion ultimately happens. Nonetheless, the average tethering time produced by Rab11 optogenetics in Exo70 KD cells was clearly longer than that produced by Exo70 optogenetics (by ~70%; Fig. 5d, black versus red line), suggesting that KS was more frequent. Repeating the Rab11 optogenetics experiments in 100 mM HEPES confirmed that prolonged tethering leads to KS (Fig. 5e, f) and, occasionally, repeated kissing (Supplementary Fig. 11). Note that vesicles that kiss a second time show a higher pre-fusion TfRc-pH signal likely because they already took up HEPES from the first kiss (Supplementary Fig. 11b).

Importantly, KS was the predominant mode of fusion with CRY2-Rab11 after Exo70 KD (Fig. 5g). Moreover, residual FF events had an average tethering time that was unusually long and not significantly different from that of KR or KS (Fig. 5f). The simplest explanation for these findings is that vesicle fusion in the absence of Exo70 is not functionally coupled to tethering by CRY2-Rab11 and thus occurs stochastically, resulting in kissing more often than FF events (see cartoon in Fig. 5h). An alternative possibility is that CRY2-CIB dimerization failed to promote FF in the context of Rab11 optogenetics, for example, due to steric reasons. However, this is unlikely for two reasons. First, Rab11

optogenetics produced tethering, in the absence of Exo70 KD, that was identical to that of Exo70 optogenetics (Fig. 5d, left traces). Second, increasing the activation intensity did not cause further inhibition of FF (Fig. 5g). Yet another explanation for its inability to rescue FF is that Rab11 optogenetics tethered vesicles outside of ideal fusion sites. However, since Exo70-KK–CRY2 also selectively induced KS under low-intensity (Fig. 4a) or low-frequency activation, (Fig. 4b–d), it is unlikely that spatial targeting by a means other than CRY2-CIB dimerization accounted for the ability of Exo70 optogenetics to promote FF. Therefore, we conclude that the exocyst actively couples tethering and fusion, both normally and optogenetically, likely by interacting with the fusion machinery[14,34].

**Full fusion but not kissing events promotes membrane expansion**. What, then, is the physiological relevance of the active promotion of FF by the exocyst? We found that Exo70 optogenetics could at times induce robust membrane expansion, particularly when cells were grown on fibronectin (Fig. 6a, b). Although tethering was stimulated using global TIRF illumination, the membrane typically expanded only on one side of the cell (Fig. 6c, asterisk). Expansion did not occur in the absence of CIB with high intensity activation (2.6 W/cm$^2$; Fig. 6c), which indicated that it was not caused by light per se. Nor did expansion occur in the presence of CIB when we used a low intensity activation (0.23 W/cm$^2$) that does not promote FF (Fig. 6b–d). We note there was an initial retraction of the membrane, which was rapid and complete within the first ~10 s (dashed box in Fig. 6b), that occurred under all optogenetic conditions (Fig. 6d) and thus likely represents a nonspecific effect of light.

Because 488-nm light was used both to image cells and to activate Exo70-KK–CRY2, it was not possible to illuminate the membrane without potentially triggering its expansion. For this reason, we labeled cells with TfRc-pHTomato, which can be excited with 561-nm light that does not activate CRY2, and we used 405-nm light instead for CRY2 activation. This allowed us to acquire a movie of the cell prior to activation and thus verify that membrane expansion is specifically triggered by Exo70 optogenetics. As expected, expansion occurred only during the activation period (Supplementary Fig. 12a and Supplementary Movie 3). Importantly, many more fusion events occurred during the two minutes right after activation (circles) than before it (crosses), particularly at the base of the expanding region (Supplementary Fig. 12b).

The above membrane expansion resembles lamellipodial protrusion induced by optogenetic activation of Rac1 (ref. [58]), a GTPase that regulates actin cytoskeletal dynamics. Accordingly, we used Lifeact-GFP[59] to visualize potential F-actin reorganization during Exo70 optogenetics. Figure 6e shows that membrane expansion indeed involved actin remodeling: pre-existing and new filopodia (arrows and arrowheads, respectively) rapidly elongated by several microns, with membrane coalescing in between them to advance the cell periphery (Supplementary Movie 4). As seen here, filopodia formation and elongation typically occurred in front of an array of focal adhesions represented by intense Lifeact-GFP puncta, suggestive of the leading edge. Membrane protrusions such as ruffles and lamellipodia have long been suggested to arise from not only the force of actin polymerization but also the circulation of membrane derived from endosomes[60,61] (i.e. TfRc-positive compartments). Collectively, our above results demonstrate that exocytosis of TfRc-containing recycling vesicles can indeed induce membrane expansion through a process that involves changes in the actin cytoskeleton.

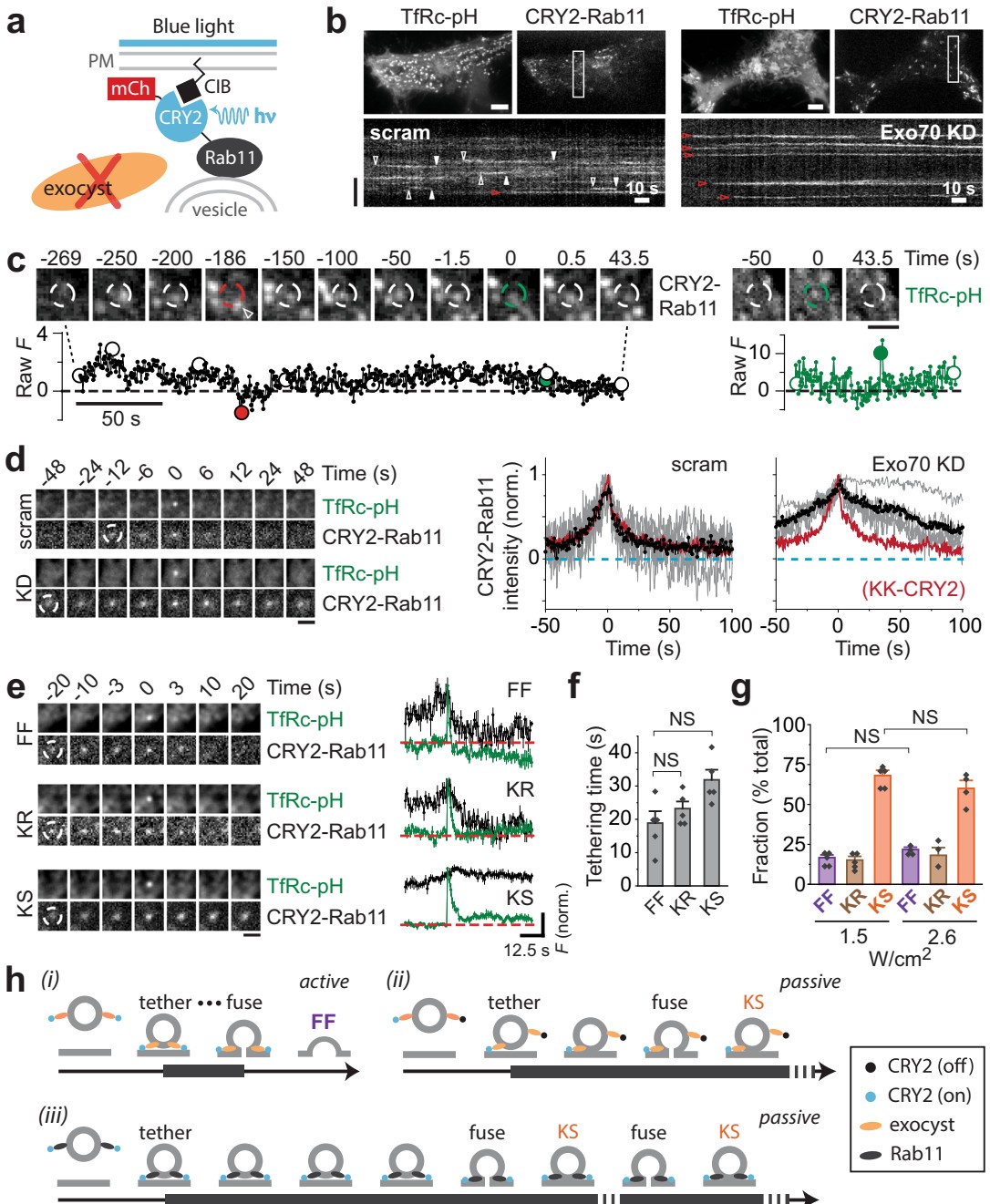

**Fig. 5 Rab11 optogenetics in Exo70 KD cells does not rescue full fusion. a** Schematic of Rab11 optogenetics using the CRY2-CIB system. **b** Rab11 optogenetics with or without Exo70 KD. Maximum-intensity projection of movies (top). Scale bars, 6 μm. Kymographs of Rab11 channel (bottom). Red arrowheads, stuck vesicles. Vertical scale bar, 6 μm. **c** Images of a single vesicle undergoing very long tethering (left, black) and fusion (right, green). Colored circles in the traces (bottom) correspond to dashed circles in the image frames (top). Scale bar, 2 μm. **d** Average image sequence of fusing vesicles from one cell (left). Scale bar, 2 μm. CRY2-Rab11 traces, time aligned to fusion (right). Averages (bold line) of cell averages (light lines) are shown. $n = 5$ cells for both scram and Exo70 KD. For comparison, the thick red line redisplays the average Exo70 trace from the Exo70-KK–CRY2 (+CIB) optogenetics experiment. Dashed cyan lines, zero baseline. **e** Three fusion modes observed with CRY2-Rab11 activation (+Exo70 KD) using 100 mM HEPES. Average image sequence of FF, KR and KS events from one cell (left). Scale bar, 2 μm. TfRc-pH (green) and CRY2-Rab11 (red) traces (average of cell averages) for all fusion modes (right). Mean ± SEM. Red dashed lines, zero baseline. **f** Tethering half-times for different fusion modes. Mean ± SEM (NS = not significant, two-tailed Student's t-test. **g** Frequency of fusion modes with Rab11 optogenetics at different light doses. Note that 1.5 W/cm² was used in **b** to **f**. Mean ± SEM (NS = not significant, two-tailed Student's t-test). **h** Schematic summary of Exo70-KK (Fig. 2) and Rab11 optogenetics results. During active tethering, optimal engagement of exocyst leads to FF (i). This model entails coupling between vesicle tethering and fusion (dots). Passive tethering occurs when the exocyst is suboptimally engaged (ii) or when vesicles are artificially tethered without the exocyst (iii). Note, for clarity, only two copies of exocyst are depicted.

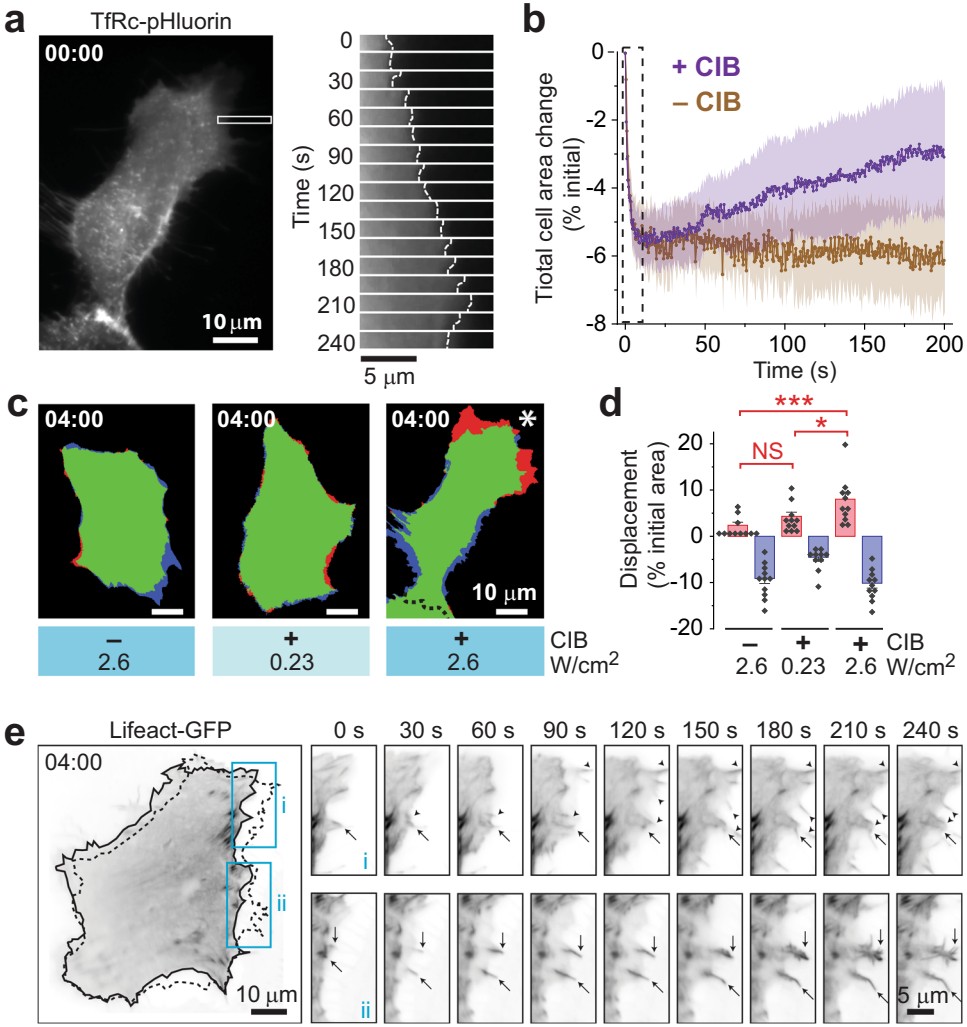

**Fig. 6 Only active tethering promotes membrane/actin cytoskeleton remodeling. a** Membrane expansion induced by Exo70 optogenetics using the CRY2-CIB system. TIRFM of TfRc-pH in a typical Exo70 optogenetics cell. A crop of a peripheral region (rectangle) shows membrane expansion (gallery, right) during activation. The cell was activated at 2 Hz with 100-ms pulses of 488-nm light (2.6 W/cm²). **b** Time-course of membrane expansion. $n = 11$ cells for both + CIB and − CIB. Mean ± SEM. Note the initial retraction caused by light during the first 5 s (dashed box). **c** Membrane expansion only occurs when active tethering is promoted in cells that express CIB optogenetic bait. Expansion (red)/retraction (blue) map of cells after 4 min. **d** Quantification of displacement. $n = 11$ cells for each condition. Mean ± SEM (*$P = 0.046$, ***$P = 0.0025$, NS = not significant, two-tailed Student's t-test). **e** Actin remodeling during Exo70 optogenetics visualized with Lifeact-GFP. Dashed outline, cell shape after 4-min activation (left). Image sequence of filopodia elongation in areas outlined by cyan rectangles (right). Arrows and arrowheads, pre-existing and new filopodia, respectively.

## Discussion

If an exocytic vesicle is merely held at the plasma membrane, will it fuse? To our knowledge, this basic question has not been addressed until the present study. In hindsight, this is surprising since it would illuminate the intrinsic efficiency of physiological fusion, which is arguably unknown. Here, by using an optogenetics approach, we were able to test the functional consequence of vesicle tethering in the absence of a native tether. First, we found that endocytic recycling vesicles undergo KR in addition to FF. Normally, recycling vesicles undergo FF more often than KR, but when exocyst function is compromised, by Exo70 KD or mutation, KR predominates. An interesting hallmark of KR observed in our study is that vesicles are tethered for a longer duration before they fuse, compared to when they undergo FF. Such a correlation between fusion mode and tethering time has been reported for synaptic vesicles[62], which suggests that it may be a general phenomenon, perhaps reflective of the mechanism of fusion (as discussed below).

Importantly, when vesicles deprived of the exocyst are tethered by light-induced dimerization of CRY2-Rab11 and CIB at the plasma membrane, they do not undergo FF. Instead, often after a long delay that can last minutes, they fuse transiently and reversibly. However, because the CRY2-CIB interaction persists, the vesicles cannot run away as they normally would and consequently remain stuck at the membrane. Thus, the longstanding question of whether passive tethering can promote fusion by increasing the probability of fusion[3] misses the point: it is not a matter of whether vesicles will fuse when passively tethered, but rather a matter of how. Our results demonstrate, that in the absence of native tethering, vesicles mainly undergo KR, and artificially tethering such vesicles to the membrane is not sufficient to promote FF.

The Exo70-KK mutant was previously shown to inhibit tethering and exocytosis by being defective in membrane binding[44]. Our live-cell imaging of single vesicles reveals that while Exo70-KK does indeed inhibit FF, it instead supports KR, suggesting that membrane binding by Exo70 is not important for

tethering per se, but rather for the mode of fusion. By controlling the membrane binding of an optogenetic analog, Exo70-KK–CRY2 (or Exo70-KK–mCherry-SspB in the iLID system), we were able to test this molecular function of Exo70 directly. Our findings strongly suggest that FF requires a tethering process in which multiple exocyst complexes bind to the membrane via Exo70. This mechanism is experimentally supported by two complementary methods of varying Exo70 engagement with the membrane. In one approach, the degree of tethering by Exo70-KK–CRY2 is controlled by modulating the intensity of light or frequency of stimulation; in another, the number of exocyst complexes available on individual vesicles (as measured by Sec8 fluorescence) is reduced by Exo70 KD. In both cases, we show that there is a high correlation between the relative number of exocyst complexes that can engage with the membrane and the mode of fusion: when Exo70-KK–CRY2 is suboptimally activated (using low light intensity or stimulation frequency), KS is selectively induced, and when fewer exocyst complexes are on vesicles, KR is favored. Altogether, our results suggest that the exocyst regulates the mode of fusion through a stoichiometric interaction of Exo70 with the plasma membrane. In support of multiple exocyst complexes being required for productive fusion, it was recently shown that approximately nine copies of the exocyst can associate with a vesicle during tethering[36], with the reported tethering duration similar to what we observed.

But why might FF require multiple exocyst complexes? One possibility is that multiple points of contact between the vesicle and the membrane are necessary to stabilize tethering for productive fusion. However, we show that varying the activation of CRY2-Rab11 does not alter the preponderance of KS, which suggests that stable tethering alone is insufficient for FF. Another possibility is that tethering and fusion are coupled through stoichiometric interactions between the exocyst and SNAREs, the components of the fusion machinery. We favor this explanation for two reasons. First, the exocyst subunit Sec3 may interact with Sso2, to facilitate formation of a binary complex with Sec9 (ref. [14]), which is the rate-limiting step in SNARE complex assembly[63], and the exocyst subunit Sec6 may interact with the assembled fusion machinery[34]. Second, the formation of a single SNARE complex may be sufficient to initiate fusion but not enough to expand a fusion pore fully[64,65]. Therefore, we speculate that multiple exocyst complexes stoichiometrically promote the formation of SNARE complexes, and perhaps spatially organize them[66], to couple tethering and FF reliably. With Exo70 acting as the molecular switch for FF, it is easy to imagine that a cell could regulate when and where a vesicle delivers its cargo by simply controlling the avidity of Exo70 for the plasma membrane, through modulation of either PI(4,5)P$_2$ (ref. [67]) or membrane-associated proteins that bind Exo70, such as the Rho family GTPase TC10 (ref. [24]).

## Methods

**Plasmids and reagents**. To generate Exo70-WT − mCherry, mouse Exo70 (GenBank accession: BC028927 [https://www.ncbi.nlm.nih.gov/nuccore/BC028927]), which is identical to human Exo70 in primary structure, was amplified by PCR and ligated into the EcoRI and KpnI sites of pmCherry-N1 (Clontech). The Exo70-KK (K632A, K625A) mutant was made by site-directed mutagenesis (QuikChange, Agilent) and validated by sequencing. To generate Exo70-CRY2 − mCherry constructs, Exo70-WT and -KK were subcloned into pCRY2PHR-mCherry[41] using NheI and XhoI sites. To generate mCherry-Rab11a, human Rab11a was subcloned into pmCherry-C1 (Clontech) using EcoRI and KpnI sites. To generate mCherry-CRY2-Rab11a, the 5-phosphatase module of OCRL in mCherry-CRY2PHR-5-ptase$_{OCRL}$[55] was replaced with Rab11a using PvuI and KpnI sites. To generate TfRc-pHTomato, pHluorin in jPA5-hTfnR-pHluorin[51] was replaced with pHTomato[52] using AgeI and XbaI sites. To generate Exo70-KK–mCherry-SspB, Exo70-KK was subcloned into a mCherry-SspB vector, which is based on tgRFPt-SspB R73Q (Micro)[68]. GFP-tagged rat Exo70 was previously described[44] and the corresponding KK mutant (GFP–Exo70-KK) was generated by site-directed mutagenesis in the same manner as Exo70-KK–mCherry. All cloning

was done using standard molecular biology techniques. Oligonucleotide primers used for molecular cloning are listed in Supplementary Table 2. For siRNA-mediated Exo70 and Sec15 KD, the following siRNA sequences were used, respectively: CCA UUG UGC GAC ACG ACU UTT and CAU GAA ACA GUU GAU GGC UAU AGA A. siRNA were purchased from Sigma.

**Cell culture and transfection**. HeLa cells (Catalog number CCL-2; ATCC) were maintained in T-75 flasks (Corning) at 37 °C and 5% CO$_2$ in DMEM (Gibco) supplemented with 4.5 g/L glucose, 1 mM sodium pyruvate, 1 × non-essential amino acids (Gibco), 10% (vol/vol) FBS and 100 U/ml penicillin-streptomycin mix (Gibco). A stable TfRc-pH cell line was generated by infecting cells with viral particles containing pLVX-puro-hTfRc-pH plasmid. Viral particle production was done in HEK293FT cells transfected with 2 μg of pLVX-puro-hTfRc-pH, 1 μg psPAX2 (Addgene, 12260), and 1 μg pMD2.G (Addgene, 12259) using Lipofecta-mine 2000 (Invitrogen, 11668-027) in DMEM, 10% FBS, without antibiotics. After overnight incubation, the medium was replaced with regular DMEM media and cells were grown for an additional 24 h. Media with virus was collected and new media was added to the transfected cells to incubate for another 24 h. The collected media was stored at 4 °C. Media after the second 24-h incubation was collected and mixed with the media previously collected and centrifuge at 500 g for 10 min to remove cell debris. The supernatant was mixed at a 3:1 ratio with Lenti-X con-centrator (Takara Bio Inc., 631231) to concentrate the virus particles after an ON incubation on ice. The mixture was centrifuged at 2000g for 1 h at 4 °C to pellet the viral particles from the media. The viral pellet was resuspended in 500 μL of PBS and 100-150 μL were used to infect HeLa cells. Positive HeLa-TfRc-pH cells were selected with 3 μg/mL puromycin after infection. For experiments with a double-stable Sec8-tagRFP/Sec8 KD cell line[35], 500 μg/ml hygromycin B (Invitrogen) and 3 μg/ml puromycin were used to select for shRNA and Sec8-tagRFP, respectively. Cells were passaged up to ~36 times and periodically checked for mycoplasma contamination.

To transiently transfect cells for live-cell imaging experiments, a Nepa21 Type II electroporator (Nepa Gene) was used according to the manufacturer's instructions. Briefly, 10$^6$ cells were resuspended in ~100 μl chilled Opti-MEM (Gibco) containing ~8 μg DNA and 16 μM siRNA (if needed) and placed into a cuvette with a 2-mm electrode gap (EC-002) and subjected to sequential trains of two and five square-voltage pulses with the following settings, respectively: (i) 125 V, 3-ms pulse length, 50-ms pulse interval, 10% decay rate and (+) polarity and (ii) 25 V, 50-ms pulse length, 50-ms pulse interval, 40% decay rate and (±) polarity. Cells were then diluted into ~12 ml phenol red-free DMEM (containing all supplements except antibiotics) and 2 ml aliquots were plated onto 35-mm glass-bottom dishes (MatTek). For membrane expansion experiments, dishes were coated with 5 mg/ml human-plasma fibronectin in PBS (Sigma-Aldrich) for >2 h prior to cell plating. Cells were imaged 2−3 days after transfection.

**Exocyst immunoprecipitation**. Stable scram, Exo70KD, Exo70-WT–mCh/Exo70 KD, and Exo70-KK–mCh/Exo70 KD HeLa cells were used for immunoprecipita-tion (IP). For each cell line a 6-cm dish was prepared with 50,000 cells/mL sus-pension. Cells were transfected with 100 nM scram or Exo70 siRNA using 7 μL of Lipofectamine RNAiMAX (Invitrogen), and incubated for 48 h. Dishes were transferred to a cold room and washed twice with 25 mM Tris-HCl pH 7.4, 20 mM NaN$_3$, 20 mM NaF, and then treated for 5 min with 350 μL of ice-cold IP lysis buffer (20 mM HEPES pH 7.4, 150 mM NaCl, 0.1% TritonX-100, 10% glycerol, 100 mM PMSF and 1 × protease inhibitor cocktail [Roche]). Cells were scraped from the dishes and transferred to 1.5-mL microcentrifuge tubes. The lysates were passed through a 25-gauge syringe six times. To clear the lysates, the samples were centrifuged for 10 min at 10,000 g at 4 °C. Supernatants were recovered and 300 μL were used for IP. The IP samples were pre-cleared for 1 h at 4 °C with 15 μL of 50% Protein A agarose beads that were pre-washed with lysis buffer. After centrifuga-tion at 600 g, the supernatants were incubated with 2 μg of mouse monoclonal against Sec15 (Kerafast) overnight with constant rotation at 4 °C. To isolate the complex, 30 μL of 50% Protein A agarose beads were added and incubated for 2 h. Samples were centrifuged at 600 g for 5 min and beads were washed three times with 400 μL of cold wash buffer (20 mM HEPES pH 7.4, 150 mM NaCl, 0.1% TritonX-100, 10% glycerol). The beads were then resuspended with 20 μL of lysis buffer and 20 μL of 2 × loading buffer and boiled for 5 min to elute the proteins from the beads. Eluents were then run on a 10% SDS-PAGE gel, transferred to nitrocellulose, and immunoblotted with antibodies against Sec15 (Kerafast, ED2003, 1:500), Exo70 (Kerafast, ED2001, 1:2000), Sec6 (Ref. [69], 1:300) and GAPDH (Cell Signaling Technology, 2118 S, 1:2000) using enhanced chemilumi-nescence (Pierce, 34580).

**Immunofluorescence**. Stable Sec8-tagRFP/Sec8 KD cells (50,000 cells/mL) were plated onto 3.5-cm glass bottom dishes (MatTek) and transfected the next day with 100 nM scram or Sec15 siRNA using 3 μL of Lipofectamine RNAiMAX per dish. The day after siRNA transfection, cells were transfected with 1.2 μg of GFP-Rab11 plasmid and 3 μL of FuGENE HD (Promega). The following day, cells were washed twice with PBS, fixed with −20 °C methanol for 10 min, and washed twice with 1 × PBS/0.05% Tween 20 (PBST). Samples were incubated for 30 min with blocking buffer (5% BSA, PBST), and incubated for 1 h with a 1:500 dilution of mouse

monoclonal Exo70 (Kerafast) in blocking buffer. Before incubation with secondary antibodies, the samples were washed three times with PBST for 5 min. The samples were incubated for 30 min with a 1:1000 dilution of goat anti-mouse labeled with Atto647N (Sigma, 50185-1ML-F) in blocking buffer, washed three times with PBST for 5 min, and then washed twice with PBS before imaging. Cells were imaged on an OMX DeltaVision V3 microscope (GE Life Sciences) equipped with 488-, 561- and 642-nm solid-state lasers (Coherent and MPB Communications), a U-PLANAPO 60×/1.42 PSF oil immersion objective lens (Olympus) and Cool-SNAP HQ$^2$ CCD cameras with a pixel size of 0.160 μm (Photometrics). The acquired wide-field illumination z-stacks images were deconvolved and aligned with SoftWorX software version 6.5.2 (Applied Precision), and the Pearson's correlation measurements were obtained with Volocity 6.3 software (PerkinElmer).

**Total Internal Reflection Fluorescence (TIRF) Microscopy**. Live-cell imaging was done using an IX-70 inverted microscope (Olympus) equipped with argon (488 nm) and argon/krypton (568 nm) laser lines (Melles Griot), a 60 × 1.45 NA oil immersion objective lens (Plan-ApoN; Olympus), and a TIRFM condenser. Cells were imaged by sequential excitation at 2 or 0.2 Hz, without binning, and detected with a back-illuminated Andor iXon887 EMCCD camera (512 × 512, 0.18 μm per pixel, 16 bits; Andor Technologies) with a 1.5× expansion lens. The TIRFM system was controlled by Andor iQ software version 1.10.1. All live-cell microscopy was done at 37 °C (using a custom incubator chamber) in phenol red-free DMEM with 10% FBS and 25 mM HEPES, pH 7.4. Prior to imaging in some experiments, 190 ml of 1 M HEPES (pH 7.4) was added to cells (in a ~2 ml volume) to obtain a final extracellular HEPES concentration of ~100 mM. Calibration of the evanescent field penetration depth was done using ~20-mm silica beads coated with fluorescent rhodamine dye as a reference object with known geometry[70,71]; the exact bead diameter was determined by taking a z-stack using a PIFOC piezo device (Physik Instrumente). For 405-nm illumination experiments, a custom TIRF microscope equipped with 405-, 488-, and 568-nm solid state lasers (Melles Griot) and a similar EMCCD camera and a 60 × 1.49 NA TIRF objective (Olympus) was used. To measure 488-nm light doses used for activation in optogenetics experiments, the laser power was measured after the objective using a S170C microscope slide power sensor (Thorlab). To determine the illumination surface area, the diameter of the laser beam was measured by capturing an image of the surface of a fluorescently labeled glass bottom dish, and this image was used to estimate the full width at half maximum of the Gaussian peak. Image frames were acquired in time-lapse recordings at 2 Hz with 150-ms exposures. Pixel size was 160 nm. For improved presentation, raw microscopy data in some figures were smoothed with a 3 × 3 pixel low pass filter using Metamorph version 7.1.2 (Universal Imaging).

**Fusion event analysis**. Vesicles undergoing fusion were identified by eye while movies were replayed. Fusion onset was defined as the first frame showing a significant fluorescence increase of the vesicle in the TfRc-pH channel. The time and location of each fusion event in a cell were noted and a 4 × 4-μm square area was centered on the brightest pixel of the vesicle and excised as a ministack for analysis. In two-color experiments, this square was transferred to the corresponding coordinates in the other-color image to produce a second ministack. Unless otherwise indicated, fluorescence of single vesicles was measured as the spatially averaged intensity difference between a 1.3-μm circle centered on the vesicle and a concentric annulus with a 1.3-μm inner and 2.4-μm outer diameter. Image analysis was done using Metamorph software version 7.1.2 (Universal Imaging).

To compute average traces of fluorescence signals, individual cell averages ($n = 3−6$) were averaged, unless otherwise indicated, in which case averages reflected pooled individual traces across cells. For TfRc-pH fluorescence, individual traces were first subtracted by the average pre-fusion intensity (last 10 s before fusion) and normalized to the maximum intensity during fusion, and then averaged for each cell. For other vesicle markers (e.g., Exo70 and Rab11), individual traces were normalized to the maximum intensity and averaged. Fusion events occurring within the first 50 frames or the last 100 frames of recordings (~480 total frames) were not computed in averages.

To measure the tethering time, the average trace of each cell (for a given vesicle marker such as Exo70) was normalized to the maximum intensity before fusion, and the intensities of timepoints between −50 s to 0 s (fusion onset) were integrated.

**Membrane expansion analysis**. Analysis of membrane boundaries was performed using MATLAB version R2014b. To identify the boundaries of the cell in each frame of a movie, image segmentation was performed on the frame by applying a simple binary threshold. The intensity threshold was determined by a semi-automated process: First, a threshold was computed automatically for each frame individually using the "graythresh" function (which uses Otsu's method) available in the MATLAB Image Processing Toolbox. If these automated thresholds did not produce a binary mask that passed visual inspection, a single threshold for all frames in the movie was chosen manually.

Once a satisfactory threshold (or set of thresholds) was found, a binary mask was produced for each frame of the movie by selecting the largest connected object in the thresholded image. In the mask, pixels with a value of 1 were considered inside the cell, and pixels with a value of 0 were considered outside the cell. Thus, the area of the cell in a given frame is computed simply by summing all pixels in the frame's mask and multiplying by the dimensions of a single pixel (i.e., the resolution of the image).

To produce images showing regions of membrane expansion and contraction, the mask of the first frame of a movie was subtracted from the mask of the last frame. In this difference image, pixels with a value of 1 indicated expansion (membrane was present in the last frame but not the first), and pixels with a value of −1 indicated contraction (membrane was present in the first frame but not the last).

**Statistics**. Statistical analyses were performed using unpaired two-tailed Student's t-tests, and $P < 0.05$ was taken as significant, indicated with asterisks (*$P < 0.05$, **$P < 0.01$, ***$P < 0.001$). Data were analyzed using Excel (Microsoft) and Origin (OriginLab).

**Reporting summary**. Further information on research design is available in the Nature Research Reporting Summary linked to this article.

## Data availability
The data that support the findings of this study are available from the authors on reasonable request. The sequence used for mouse Exo70 is available at GenBank under Accession Code BC028927.1 [https://www.ncbi.nlm.nih.gov/nuccore/BC028927]. Source data are provided with this paper.

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

## Acknowledgements

We thank J. Bogan, C. Burd, P. De Camilli, J. Rothman and D. Zenisek for comments on the manuscript, and I. Kukic and L. Watson for help with the preparation of the manuscript. Supported by the National Institutes of Health R01GM098498 and R01GM118486. The data are presented in the main manuscript and the supplementary materials.

## Author contributions

S.J.A., F.R.-M., and D.T. conceived the project. S.J.A. and F.R.-M. performed experiments and data analysis. A.A. made the Exo70-KK version of GFP-Exo70 and performed confocal microscopy experiments. Z.X. made the iLID optogenetics constructs. B.M. and V.I.P. provided technical expertise. S.J.A. and D.T. wrote the manuscript.

## Competing interests

The authors declare no competing interests.

**Additional information**

