## [Peer Review File · Nature Communications]

Reviewers' comments:

Reviewer #1 (Remarks to the Author):

This new manuscript from the Toomre lab is focused on identifying the role(s) of the mammalian exocyst complex in secretory vesicle tethering and membrane fusion. To do this, they compared tethering/fusion of vesicles using a knock-down of the exocyst component Exo70, or a mutant Exo70KK, and an exocyst complex optogenetically brought to the plasma membrane for fusion, vs. an artificial tether (optogenetically tethered Rab11). They observed differences in the fusion of vesicles under these situations, suggesting that impairment of exocyst, or use of the artificial tether, leads to vesicles using kiss-and-run (or even kiss-and-stay) fusion. These data indicate a critical role for exocyst in facilitating proper physiological fusion events.

The experiments are very interesting, and the use of a clever optogenetic trigger to bring exocyst to the membrane, as compared to degrading/knockdown of it. However, there are a number of underlying assumptions (outlined below) that may or may not support the experimental conclusions. In a number of cases, the figure legends really do not contain enough detailed information for the figure to be understood without thorough cross-referencing the main text (and that did not always help either).

It is unclear what vesicles are actually being followed in these experiments. The authors refer to them as recycling vesicles (presumably that assumption is based on the presence of Rab11?), but shouldn't exocyst subunits be on exocytic vesicles too? Are there multiple populations being examined, and do we expect them to all have the same kinetics of tethering and fusion?

The use of the Exo70KK mutant is based on data from another group (Liu et al, MBoC 2007) that showed very different interaction of Exo70 (and presumably intact exocyst complexes) at the plasma membrane, which the authors here do not observe. How might this discrepancy be reconciled? The loss of Exo70 (exocyst) at the plasma membrane with this KK mutant is not definitively shown in the authors' system, although most of the conclusions using this mutant assume that it is defective for membrane interaction. Is it possible that the presence of Exo70 (wt) affects exocyst function in another way besides membrane recruitment, e.g. the mutants destabilize exocyst complexes, although they may still be assembled? Along those lines, the IP experiment in Suppl Fig 1 seems to suggest Sec15 and Sec6 assembling together in the Exo70 KD with no Exo70 present (presumably there is some below the detection level of that blot?)? This experiment should be quantified, which would also help indicate the amount of KD of Exo70 in various experiments? Lack of quantification in several experiments is similarly problematic.

The data supporting kiss-and-run is not compelling enough. That conclusion is mainly based on minor differences in the intensity of the fluorescence when in 100 mM Hepes buffer, and a slightly extended kinetics. Another orthogonal technique would really help the authors conclude that they are observing kiss-and-run (or kiss-and-stay). Similarly, another assumption, which might be valid but is untested, is that spots of Exo70 or Sec8 at the membrane are actually tethered/tethering vesicles.

However, I would agree that something different is going on with "fusion" when exocyst is compromised by the KK mutants of Exo70, but it's a leap to identify it as kiss-and-run. Kiss-and-run has clearly been demonstrated and studied in neuronal cells, in which fusion is Ca²⁺-dependent and

potentially reversible, but not clearly defined for HeLa cells. Are there other mechanistic explanations that would account for the observations that this mutant still appears to tether vesicles, but alters what appears to be “fusion”?

The authors indicate (p. 9) that the 488 nm light is used for both TfRc-pH imaging and activation of CRY2, but do not really explain how the experiment is performed such that this problem does not affect the experimental outcome, especially as they show that increased frequency of CRY2 activation affects the “fusion” state of the vesicles.

In terms of the CRY2 activation, is there a way to distinguish the number of exocyst complexes at the fusion site, compared to w.t. exocyst or in the Exo70 KD?

For the artificial optogenetic Rab11 experiment, is the endogenous Rab11 knocked down? How overexpressed is the optogenetically tagged Rab11, and is it possible that there is more (or less) functional Rab11 on vesicles? That could explain many of the observed results with that construct (e.g. stuck vesicles).

With regard to the model and the discussion, another confusing aspect is how to understand this mechanistically, as the SNAREs themselves (which are doing the fusion) are unchanged, only exocyst is. Really need a better mechanistic model for exocyst function (or lack thereof), the speculative discussion is insufficient to support the conclusions drawn.

Reviewer #2 (Remarks to the Author):

In this study, An et al. suggest the active tethering mechanism of vesicle fusion control the fate of vesicles mainly using TRIF microscopy and pH-sensitive exocytosis reporter (TfRc-pH). They showed active engagement between tethering and fusion in the role of exocyst with the advantage of Exo70-KK, Lysin mutation of Exo70 mutation blocking delivery of secretory proteins on the cell surface. To verify their suggestion, they recruited PHR-CIB heterodimerization module, which demonstrated the reasonable active formation of full fusion of vesicle. Nevertheless, as the authors mentioned, it is compromised that tracking the exocyst and its vesicle compartment when using PHR-CIB and pHluorin in the same field of imaging. Last, the author suggests the only active tethering promotes reassemble of cellular structure, actin cytoskeleton. Here, I have my comments below. Most importantly, the authors should show the same effects with other optogenetic dimerization systems, such as iLID due to CRY2 oligomerization properties on membranes.

1. To compare between effects on cellular functions such as membrane tethering and fusion using optogenetics tool the laser intensity and frequency was important. However, the author compared different Hz of light stimulation in fig.3d; different light intensity was used in fig.5c,d. That should be compared with the same light frequency and intensity.
2. It is not precisely what conditions of light stimulation were tested. Also, It is also necessary to describe the total amount of stimulated light.
3. It may not be the result of exo70 pulled into the membrane, but excessive CIB localizes on the membrane by CRY2 in fig.3, 5. It is necessary to stimulate light to cells just expressing cry2-mCherry

that was not tagged at exo70.

4. Why does full fusion continue only at the edge of the pre-cell, rather than in the region of the post-cell?
5. Are there any results that can be controlled by varying intensity or frequency conditions to control the fate of vesicles? If it is available, it could be applied to a wider range of studies.
6. CRY2 tends to aggregate on membranes (PM, ER, intracellular membranes) and induce aggregation of endosomes and vesicles even in the dark condition. The authors SHOULD show the same effects with other optogenetic dimerization systems, such as iLID or Magnet systems.

Reviewer #3 (Remarks to the Author):

In this manuscript, An et al. characterize the impact of the exocyst complex on the mode of vesicle fusion. Using optogenetic approaches in HeLa cells, the authors propose that the exocyst complex not only promotes vesicle docking, but also plays a decisive role in the outcome of vesicle fusion. They report that the exocyst facilitates full fusion over kiss-and-run, invoking the idea that SNAREs alone are not sufficient to catalyze full fusion in living cells. In addition, full fusion regulated by the exocyst is important to regulate, in turn, membrane expansion. Modern approaches are used throughout this study, and new findings reported here shed new light on our understanding of the exocyst complex. I have considerable enthusiasm for this study. However, a few key issues need to be addressed, as listed below in no specific order.

1. Why use HeLa cells to study vesicle exocytosis? One might expect to see this kind of work done in cells where release can be triggered by Ca^{2+} , as in neuroendocrine cells. In such cases, there are docked pools of vesicles; are there docked pools in HeLa cells? Or, in HeLa cells is all fusion constitutive, so docked pools never formed? This reviewer is thinking out loud here, as I was surprised to see the HeLa cells were the model system, so some rationale for this choice might be constructive.
2. This reviewer cannot understand how the data in Figure 5 reveal that vesicle fusion contributes to membrane expansion. It might be helpful if the authors can provide evidence that active vesicle exocytosis is taking place at the site of membrane expansion.
3. It is well recognized that the exocyst complex helps to tether and dock vesicles. However, this is not studied at all in this manuscript. In particular, the Exo70-KK mutant should significantly decrease vesicle docking. It seems that it would be important to examine docking in this study.
4. Why use 100 mM HEPES to determine the mode of fusion? In other studies, researchers use 5 versus 25 mM HEPES, so it is not clear why the authors of the current study use 25 and 100 mM HEPES.

The authors cite ref.37 for this assay. However, ref. 37 is a review from Richard Tsien; the original paper should be cited.

In addition, the pHluorin traces in full fusion events in this manuscript are very different from Ref.

37. The authors should explain this difference. Moreover, to my eye, the decay in Fig. 1F is faster in 100 mM HEPES, not slower. To be clear: on Page 8, the authors write, “100 mM HEPES affected TfRc-pH fluorescence only with Class ii events, ablating the faster decay of the receptor compared to ligand in 25 mM HEPES (dashed boxes in Fig. 1f)”. However, again, Fig. 1f shows that decay of the receptor is faster than ligand using 100 mM HEPES.

5. The authors need to edit the paper more carefully. Quite a few sentences are difficult to understand. For example, on Page 7, the authors wrote, “With TfRc-pHTomato, the tethering duration was similar to that observed with Exo70-WT, Sec8 and Rab11, but when Exo70 KD was knocked down, the duration was anomalously long.” What is Exo70 KD?

Overall, there are interesting results in the manuscript, but it is a bit difficult to understand some of the findings.

Note: New experiments and analysis performed for the revision are indicated in **boldface**.

Reviewer #1 (Remarks to the Author):

This new manuscript from the Toomre lab is focused on identifying the role(s) of the mammalian exocyst complex in secretory vesicle tethering and membrane fusion. To do this, they compared tethering/fusion of vesicles using a knock-down of the exocyst component Exo70, or a mutant Exo70KK, and an exocyst complex optogenetically brought to the plasma membrane for fusion, vs. an artificial tether (optogenetically tethered Rab11). They observed differences in the fusion of vesicles under these situations, suggesting that impairment of exocyst, or use of the artificial tether, leads to vesicles using kiss-and-run (or even kiss-and-stay) fusion. These data indicate a critical role for exocyst in facilitating proper physiological fusion events.

The experiments are very interesting, and the use of a clever optogenetic trigger to bring exocyst to the membrane, as compared to degrading/knockdown of it. However, there are a number of underlying assumptions (outlined below) that may or may not support the experimental conclusions. In a number of cases, the figure legends really do not contain enough detailed information for the figure to be understood without thorough cross-referencing the main text (and that did not always help either).

We thank the reviewer for pointing out the value of acute manipulations (i.e., optogenetics) versus chronic degradation/knockdown to study the exocyst. We address below key assumptions and new data to support our conclusions. We agree that more details were needed in the figures and now provide additional details in the figure legends to clarify the experimental conditions; these were previously left out due to space limitations.

(1) It is unclear what vesicles are actually being followed in these experiments. The authors refer to them as recycling vesicles (presumably that assumption is based on the presence of Rab11?), but shouldn't exocyst subunits be on exocytic vesicles too? Are there multiple populations being examined, and do we expect them to all have the same kinetics of tethering and fusion?

This is an important point, and we thank the reviewer for giving us an opportunity to discuss it. We fully agree that exocytic vesicles can arise from compartments other than the recycling compartment – namely, the Golgi/TGN (Rodriguez-Boulan E, Kreitzer G. & Musch A. *Nature Rev. Mol. Cell Biol.* 6, 233–247 (2005).

However, several pieces of evidence support our interpretation that the pool of vesicles engaged by the exocyst are from a recycling compartment:

1) We have previously shown that vesicles labeled with post-Golgi markers such as VSVG and NPY do not substantially colocalize with the exocyst (Rivera-Molina F & Toomre D, *J Cell Biol.* 201: 673-680, 2013). This strongly indicates that there are distinct sets of exocytic vesicles – one that is exocyst dependent and another that is exocyst independent.

2) Post-Golgi vesicles labeled with hGH were reported to undergo mainly kiss-and-run fusion (Jaiswal JK, Rivera VM & Simon SM, *Cell* 137:1308–19, 2009), which may indeed reflect the absence of exocyst on post-Golgi vesicles.

3) We note that in our study, Rab11-positive vesicles have tethering times that are very similar to those of Exo70- and Sec8-positive vesicles (Supplementary Fig. 4d). This argues that Rab11 and exocyst subunits are labeling the same population of vesicles.

For these reasons, when we follow transferrin receptor to monitor exocytosis, it is unlikely that we are examining multiple populations of exocytic vesicles (i.e., *both recycling and post-Golgi vesicles*).

(2) The use of the Exo70KK mutant is based on data from another group (Liu et al, *MBoC* 2007) that showed very different interaction of Exo70 (and presumably intact exocyst complexes) at the plasma membrane, which the authors here do not observe. How might this discrepancy be reconciled? The loss of Exo70 (exocyst) at the plasma membrane with this KK mutant is not definitively shown in the authors' system, although most of the conclusions using this mutant assume that it is defective for membrane interaction.

We agree with the noted discrepancy between our findings of Exo70 localization and those of the Liu et al paper. The major difference was that Liu et al. overexpressed Exo70-WT and Exo70-KK without knockdown of endogenous Exo70. Thus, most of their overexpressed Exo70 could not be in the exocyst complex, and the presumed differential targeting was driven by the residues K632 and K635 present in Exo70-WT but not Exo70-KK.

To reconcile this and directly demonstrate that Exo70-KK is defective in membrane binding, as suggested by the reviewer, we have now imaged overexpressed Exo70-WT and -KK without knockdown of endogenous Exo70 (i.e., without molecular replacement) by confocal microscopy and confirmed that Exo70-WT but not -KK binds to the plasma membrane (Supplementary Fig. 1). Our new confocal data are comparable to the original paper demonstrating the membrane-binding deficiency of Exo70-KK (Liu J. et al. *Mol. Biol. Cell* 18: 4483–4492, 2007). Furthermore, they support the idea that when Exo70 is over-expressed, the overexpressed subunit can be driven to the membrane independent of the exocyst, as the localization patterns of overexpressed tagged Exo70 with and without molecular replacement are spatially distinct.

(3) Is it possible that the presence of Exo70 (wt) affects exocyst function in another way besides membrane recruitment, e.g. the mutants destabilize exocyst complexes, although they may still be assembled? Along those lines, the IP experiment in Suppl Fig 1 seems to suggest Sec15 and Sec6 assembling together in the Exo70 KD with no Exo70 present (presumably there is some below the detection level of that blot?)? This experiment should be quantified, which would also help indicate the amount of KD of Exo70 in various experiments? Lack of quantification in several experiments is similarly problematic.

The point is a fair one and we believe that the most straightforward explanation for the functional difference between Exo70-WT and the KK mutant is that the latter is defective in membrane binding (Liu J. et al. *Mol. Biol. Cell* 18: 4483–4492, 2007) – **a feature of Exo70-KK that we have now verified (Supplementary Fig. 1).** Furthermore, we note the while the KK mutant is unable to support full fusion, full fusion can be rescued by optogenetically promoting membrane binding of the KK mutant (i.e., by using Exo70-KK-CRY2). Since our optogenetic Exo70 construct also carries the KK mutation, it is unlikely

that the KK mutant indirectly affects fusion in other ways, for example, by somehow destabilizing the exocyst complex.

Regarding the Sec15 IP experiment in Supplementary Fig. 2, we have quantified the pulldown of the exocyst subunits in the below graph. We note that this experiment also does not support the idea that the KK mutant destabilizes exocyst complexes. Furthermore, we have also quantified the intensity of Exo70 WT and the KK mutant on vesicles (see below), and we find that there is no significant difference in intensity between Exo70-WT and-KK, which again argues against the KK mutant destabilizing exocyst complexes.

(4) The data supporting kiss-and-run is not compelling enough. That conclusion is mainly based on minor differences in the intensity of the fluorescence when in 100 mM Hepes buffer, and a slightly extended kinetics. Another orthogonal technique would really help the authors conclude that they are observing kiss-and-run (or kiss-and-stay).

We appreciate the comment and **have now used an orthogonal technique to detect kiss-and-run fusion. We have performed radial line scan measurements of vesicles labeled with transferrin-Alexa568 undergoing fusion. Line scans were calculated for consecutive images before and during fusion and fitted by Gaussian functions to discern full fusion, as described (Zenisek D, et al. *Neuron* 35: 1085–1097, 2002). As shown in Supplementary Figure 7, full fusion clearly shows lateral spread of transferrin-Alexa568 based on the increase of the width of the Gaussian curves after fusion. On the other hand, kiss-and-run does not show lateral spread, as the full width half maximum remains the same over time.**

Thus, we have now used two orthogonal techniques to support the presence of kiss-and-run fusion – one based on the sensitivity of a pH-sensitive fusion reporter to external buffers (i.e., 100 mM HEPES) and one based on radial sweeps of pH-insensitive vesicle cargo before and during fusion.

(5) Similarly, another assumption, which might be valid but is untested, is that spots of Exo70 or Sec8 at the membrane are actually tethered/tethering vesicles.

This is an excellent point and one that we believe is particularly relevant when exocyst subunits are highly overexpressed (as discussed above for Exo70). Indeed, we have previously shown that when Sec8 is overexpressed *without* knockdown of the endogenous protein, very large aggregates of Sec8 can form (Rivera-Molina F & Toomre D, *J Cell Biol.* 201: 673-680, 2013). As such, we were careful to express

exocyst subunits at low levels *and* only with knockdown of endogenous subunits to favor molecular replacement of exocyst subunits.

In the present study, two additional tests support the idea that Exo70 and Sec8 spots at the membrane are tethered/tethering vesicles. First, Exo70 and Sec8-tagRFP spots colocalize with Rab11 spots in a Sec15-dependent manner (Fig. 1a). Second, the number (as well as the intensity) of Sec8 spots is reduced when the exocyst complex is destabilized by Exo70 knockdown (Fig. 4e). These two findings further argue that dim, dynamic Exo70 and Sec8 spots produced by concomitant knockdown of Exo70 or Sec8 are not merely protein aggregates.

(6) However, I would agree that something different is going on with “fusion” when exocyst is compromised by the KK mutants of Exo70, but it’s a leap to identify it as kiss-and-run. Kiss-and-run has clearly been demonstrated and studied in neuronal cells, in which fusion is Ca²⁺-dependent and potentially reversible, but not clearly defined for HeLa cells. Are there other mechanistic explanations that would account for the observations that this mutant still appears to tether vesicles, but alters what appears to be “fusion”?

We agree that kiss-and-run has been studied extensively for Ca²⁺-dependent synaptic vesicles exocytosis in neurons (Aravanis AM, Pyle JL & Tsien RW, *Nature* 423: 643–647, 2003; Gandhi SP & Stevens CF, *Nature* 423: 607–613, 2003). However, whether synaptic vesicles undergo kiss-and-run is much debated because these vesicles are small (~30 nm) and thus challenging to detect by light microscopy. On the other hand, exocytic vesicles in non-neuronal cells are much larger (> 100 nm). As such, microscopic evidence for kiss-and-run is arguably clearer for vesicles in non-neuronal cells. For example, kiss-and-run has been visualized compellingly in at least three types of non-neuronal cells: i) PC12 cells (Holroyd P, Lang T, Wenzel D, De Camilli P & Jahn R, *Proc. Natl Acad. Sci. USA* 99: 16806–16811, 2002), ii) lactotrophs (Vardjan N et al. *J. Neurosci.*, 27: 4737–4746, 2007) and iii) pancreatic islet beta-cells (Tsuboi T & Rutter GA, *Biochem. Soc. Trans.*, 31: 833–836, 2003). The evidence for kiss-and-run fusion in non-neuronal cells is not limited to imaging as it has been strongly supported by electrophysiological methods (Albillos, A et al. *Nature* 389: 509–512, 1997; Alés E. et al. *Nat. Cell. Biol.* 1: 40–44, 1999; Wang CT et al. *Nature* 424: 943–947, 2003).

But to the point of whether kiss-and-run exocytosis can occur in HeLa cells, a constitutive kiss-and-run pathway has been demonstrated in HeLa cells (Okayama M, et al. *Cell Struct. Funct.* 34:115–125, 2009). Furthermore, a study from Simon and colleagues convincingly showed that post-Golgi vesicles in HT1080 cells (i.e., another fibroblastic human cell line like HeLa cells) undergo kiss-and-run, in a manner that is completely independent of Ca²⁺ (Jaiswal JK, Rivera VM & Simon SM, *Cell* 137:1308–19, 2009).

Thus, we believe there is strong evidence for kiss-and-run in non-neuronal cells, for both Ca²⁺-dependent and -independent forms of exocytosis, but we also recognize that this literature may not be obvious as most discussion of kiss-and-run has been in neurons, and as such we have added a statement and citation describing the existence of kiss-and-run in non-neuronal cells.

(7) The authors indicate (p. 9) that the 488 nm light is used for both TfRc-pH imaging and activation of CRY2, but do not really explain how the experiment is performed such that this problem does not affect the experimental outcome, especially as they show that increased frequency of CRY2 activation affects the “fusion” state of the vesicles.

In our optogenetics experiments, we used 488-nm light both to image transferrin-receptor-pHluorin (TfRc-pH) and to activate CRY2, since both pHluorin and CRY2 absorb blue light. Our acquisition/activation rate of 2Hz is sufficient to image fusion events and activate exocytosis optogenetically. The only complication of this system is that exocytosis cannot be imaged without concurrently activating CRY2 when TfRc-pH is used to monitor exocytosis. As described in the manuscript (Fig. 6e), we overcame this limitation by monitoring exocytosis instead with TfRc-pHTomato, which can be excited with 550-nm light (i.e., a wavelength that does not activate CRY2).

(8) In terms of the CRY2 activation, is there a way to distinguish the number of exocyst complexes at the fusion site, compared to w.t. exocyst or in the Exo70 KD?

(9) For the artificial optogenetic Rab11 experiment, is the endogenous Rab11 knocked down? How overexpressed is the optogenetically tagged Rab11, and is it possible that there is more (or less) functional Rab11 on vesicles? That could explain many of the observed results with that construct (e.g. stuck vesicles).

These are good points, and we can address these two questions together. **The graph below shows the intensities of different Exo70 and Rab11 proteins (all tagged with mCherry) on vesicles.** Notably, the amounts of exogenously expressed Exo70 and Rab11 on vesicles are similar across all experiments, including those of optogenetically tagged Exo70 and Rab11. The similarity in intensity of Exo70-KK-CRY2 and CRY2-Rab11 indicates that the functional difference between the two constructs does not reflect the level of expression of these proteins. Furthermore, we note that the promotion of stuck vesicles is not an intrinsic feature of Rab11 optogenetics for two reasons: 1) CRY2-Rab11 does not produce stuck vesicles when Exo70 is *not* knocked down (Fig. 5d), and 2) stuck vesicles are also promoted in optogenetic Exo70 experiments when Exo70-KK-CRY2 is activated with low frequency stimulation (Fig. 4b-d).

(10) With regard to the model and the discussion, another confusing aspect is how to understand this mechanistically, as the SNAREs themselves (which are doing the fusion) are unchanged, only exocyst is. Really need a better mechanistic model for exocyst function (or lack thereof), the speculative discussion is insufficient to support the conclusions drawn.

Rothman and colleagues have demonstrated that reversible fusion can be reconstituted *in vitro* with lipids containing purified SNARE proteins alone (Shi L, et al. *Science* 335: 1355–1359, 2012) – without the addition of any other regulators of exocytosis. Specifically, they showed that the fusion pore reseals rapidly when fewer than three SNAREpins assemble, which indicates that several SNAREpins are needed to provide the radial force to fully fuse vesicles. A similar finding – that full fusion (specifically, content mixing versus lipid mixing in reconstituted fusion assays) is highly sensitive to the number of SNAREpins – has been reported by Chapman and colleagues (Bao H, et al. *Nat. Struct. Mol. Biol.* 23: 67–73, 2016). Given that the exocyst promotes formation of the binary t-SNARE complex (Yue P, et al. *Nat. Commun.* 8: 14236, 2017), and also interacts with assembled SNAREpins (Dubuke ML, et al. *J. Biol. Chem.* 290:28245–28256, 2015), we believe that our model of how the exocyst controls fusion mode – by stoichiometrically promoting SNAREpin assembly – is reasonable, particularly in the context of a ring of exocyst complexes around the interface of the vesicle and the plasma membrane, based on the *in vivo* architecture of the exocyst complex (Picco, A. et al. *Cell* 168, 400–412.e18, 2017), and with consideration of the “buttressed-ring” hypothesis by Rothman and colleagues (Rothman JE, et al. *FEBS Lett.* 591, 3459–3480, 2017), which proposes a stoichiometric coordination of accessory proteins organized as a ring with SNARE complexes to provide the precise topology of SNAREpins for efficient membrane fusion.

Reviewer #2 (Remarks to the Author):

In this study, An et al. suggest the active tethering mechanism of vesicle fusion control the fate of vesicles mainly using TRIF microscopy and pH-sensitive exocytosis reporter (TfRc-pH). They showed active engagement between tethering and fusion in the role of exocyst with the advantage of Exo70-KK, Lysin mutation of Exo70 mutation blocking delivery of secretory proteins on the cell surface. To verify their suggestion, they recruited PHR-CIB heterodimerization module, which demonstrated the reasonable active formation of full fusion of vesicle. Nevertheless, as the authors mentioned, it is compromised that tracking the exocyst and its vesicle compartment when using PHR-CIB and pHluorin in the same field of imaging. Last, the author suggests the only active tethering promotes reassemble of cellular structure, actin cytoskeleton. Here, I have my comments below. Most importantly, the authors should show the same effects with other optogenetic dimerization systems, such as iLID due to CRY2 oligomerization properties on membranes.

We thank the reviewer for recommending a second optogenetic dimerization system. **As described below, we have now recapitulated our main findings using the iLID system, which rules out the possibility that CRY2 oligomerization influences optogenetic-induced tethering in the CRY2(PHR)/CIB system. We present this new data as its own main figure (Fig. 3) since we feel that it greatly strengthens the manuscript.**

1. To compare between effects on cellular functions such as membrane tethering and fusion using optogenetics tool the laser intensity and frequency was important. However, the author compared different Hz of light stimulation in fig.3d; different light intensity was used in fig.5c,d. That should be compared with the same light frequency and intensity.

We apologize for any confusion caused by not always specifying the precise light conditions used to stimulate cells (see next comment). In fact, the *same* light conditions (frequency and intensity) were

used to promote vesicle tethering (Fig. 2) and membrane expansion (Fig. 6c,d). Specifically, to induce membrane expansion, we used a light intensity and frequency (2.6 W/cm^2 and 2 Hz) that *maximally* promotes full fusion, based on the dose-response relationship between light intensity and fusion mode described in Figure 4a. We should note – that to rule out a nonspecific effect of light on membrane expansion under these conditions – we also tested cells that did not express the optogenetic bait (i.e., minus-CIB condition). In cells without CIB, maximal stimulation (2.6 W/cm^2 , 2Hz) neither promoted full fusion (Fig. 2a) nor induced membrane expansion (Fig. 6c,d). Thus, we can be confident that membrane expansion was specifically induced by the optogenetic promotion of full fusion.

2. It is not precisely what conditions of light stimulation were tested. Also, It is also necessary to describe the total amount of stimulated light.

We have now specified the total amount of light used to stimulate cells wherever it was missing in the text or figure legends. Again, we apologize for any confusion this may have caused.

3. It may not be the result of exo70 pulled into the membrane, but excessive CIB localizes on the membrane by CRY2 in fig.3, 5. It is necessary to stimulate light to cells just expressing cry2-mCherry that was not tagged at exo70.

We agree that it is necessary to demonstrate that the effects of Exo70 optogenetics do not reflect some minor aspect of the CRY2-CIB interaction itself. As the reviewer suggests, it is possible that a local enrichment of CIB (i.e., the bait) near the vesicle, caused by activation of CRY2 (i.e., the prey/photoreceptor) on the vesicle, could affect vesicle fusion. For this reason, we tested CRY2 attached to Rab11 instead of Exo70 (i.e., Rab optogenetics). As described in the manuscript, Rab11 optogenetics did not promote full fusion (Fig. 5). Thus, it is unlikely that the promotion of full fusion by Exo70 optogenetics reflected the CRY2-CIB interaction per se. **Furthermore, based on the suggestion of the reviewer, we have recapitulated our main findings using the iLID system (see below), which is a wholly independent dimerization system.** For example, in the iLID system, the prey/photoreceptor is localized to the plasma membrane, not the vesicle (which is the opposite of the CRY2/CIB system). Thus, it is unlikely that local enrichment of an optogenetic bait on the plasma membrane by a prey/photoreceptor on the vesicle is responsible for the promotion of full fusion in our optogenetics experiments.

4. Why does full fusion continue only at the edge of the pre-cell, rather than in the region of the post-cell?

The membrane expansion we observe likely represents lamellipodial protrusion, given 1) its resemblance to membrane expansion induced by optogenetic Rac1 activation in mouse embryonic fibroblasts as well as in HeLa cells (Wu Y, et al. *Nature* 461: 104–108, 2009), and 2) the concurrence of actin remodeling during expansion (i.e., filopodial formation/elongation; Fig. 6g). **We have imaged paxillin (i.e., a focal adhesion marker) during optogenetically-induced membrane expansion and find that nascent adhesions form immediately behind the leading edge (see below figure) – again, consistent with lamellipodial protrusion (Parsons JT, Horwitz AR & Schwartz MA. *Nature Rev. Mol. Cell Biol.* 11: 633–643, 2010).** Since lamellipodia are known to be very flat (~150 nm thick) and enriched in actin filaments (Abercrombie M, Heaysman JE & Pegrum SM. *Exp. Cell Res.* 67:359–367, 1971), we posit that recycling vesicles (>100 nm diameter) might be physically excluded from the expanding region,

particularly near the leading edge – explaining why fusion is not observed in the expanding region post-stimulation (Fig. 6f).

5. Are there any results that can be controlled by varying intensity or frequency conditions to control the fate of vesicles? If it is available, it could be applied to a wider range of studies.

As we describe in the Introduction, the exocyst is implicated in a myriad of biological processes (Heider MR & Munson M. *Traffic* 13: 898–907, 2012), which makes our experimental approach potentially applicable to a wide range of studies.

6. CRY2 tends to aggregate on membranes (PM, ER, intracellular membranes) and induce aggregation of endosomes and vesicles even in the dark condition. The authors SHOULD show the same effects with other optogenetic dimerization systems, such as iLID or Magnet systems.

As mentioned above, we have now used another optogenetic dimerization system – the iLID system – to test the validity of our main findings. As the referee may imagine, this new set of experiments represented a considerable amount of work (and took a while to complete), but we truly appreciate the referee’s suggestion since using an independent optogenetic technique greatly strengthened our conclusions.

As shown in a new main figure (Fig. 3), activation of vesicle tethering by using Exo70-KK–SspB and iLID-CAAX promoted full fusion, in a manner similar to that with Exo70-CRY2 and CIB-CAAX.

Importantly, when the photoreceptor (iLID-CAAX) was not co-expressed, vesicles carrying Exo70-KK–SspB did not undergo full fusion. Furthermore, the correlation between tethering time and fusion mode (i.e. short and long tethering times for full fusion and kiss-and-run, respectively) was also observed with the iLID system, which indicates that this correlation is robust. Given that synaptic vesicles show the same correlation between tethering time and fusion mode (Park HY & Tsien RW. *Science* 335: 1362–1366, 2012) but do not depend on the exocyst for tethering (Murthy M. et al, *Neuron* 37: 433–447, 2003), it is reasonable to argue that kiss-and-run is a basic feature of SNARE-mediated fusion, as we discuss in our conclusion.

Reviewer #3 (Remarks to the Author):

In this manuscript, An et al. characterize the impact of the exocyst complex on the mode of vesicle fusion. Using optogenetic approaches in HeLa cells, the authors propose that the exocyst complex not only promotes vesicle docking, but also plays a decisive role in the outcome of vesicle fusion. They report that the exocyst facilitates full fusion over kiss-and-run, invoking the idea that SNAREs alone are not sufficient to catalyze full fusion in living cells. In addition, full fusion regulated by the exocyst is important to regulate, in turn, membrane expansion. Modern approaches are used throughout this study, and new findings reported here shed new light on our understanding of the exocyst complex. I have considerable enthusiasm for this study. However, a few key issues need to be addressed, as listed below in no specific order.

We thank the reviewer for the thoughtful summary of our work and expressing enthusiasm for it.

1. Why use HeLa cells to study vesicle exocytosis? One might expect to see this kind of work done in cells where release can be triggered by Ca^{2+} , as in neuroendocrine cells. In such cases, there are docked pools of vesicles; are there docked pools in HeLa cells? Or, in HeLa cells is all fusion constitutive, so docked pools never formed? This reviewer is thinking out loud here, as I was surprised to see the HeLa cells were the model system, so some rationale for this choice might be constructive.

This is a great question, and we thank the reviewer for giving us an opportunity to discuss the finer points of vesicle docking. Regarding whether docked vesicles exist in HeLa cells, exocytosis in HeLa cells has been shown to be Ca^{2+} -independent in at least two papers (Jaiswal JK, Andrews NW & Simon SM, *J. Cell Biol.* 159: 625–635, 2002; Jaiswal JK, Rivera VM & Simon SM, *Cell* 137:1308–19, 2009). Furthermore, we observe fusion with only newly arrived vesicles, no matter how we label vesicles (e.g. fluorescent exocyst subunits, Rab11, transferrin-Alexa568 or transferrin-receptor-pHTomato). Thus, we do not believe that HeLa cells have a docked pool of recycling vesicles, nor that there would be a reason for docking in these cells, as exocytosis is constitutive.

The initial goal of our study was to answer a simple question: If we could optogenetically tether vesicles, would they fuse? The answer to this was unclear. As the reviewer points out, exocytosis has been largely studied in the context of Ca^{2+} -dependent secretion, such as in neuroendocrine cells. In such systems, secretory vesicles undergo docking, priming and finally Ca^{2+} -triggered fusion. Docking is defined morphologically, based on the physical proximity of a vesicle with the plasma membrane, as determined by electron microscopy (reviewed in Verhage M & Sørensen JB, *Traffic* 9: 1414–1424, 2008). On the other hand, priming is a biochemical process involving multiple reactions (Parsons TD et al. *Neuron* 15: 1085–1096, 1995), likely including pre-assembly of the SNARE complex to some degree (Xu T et al. *Cell* 99: 713–722, 1999). As such, in neuroendocrine cells, vesicles do not immediately fuse after they undergo tethering and even priming.

In HeLa cells – which possess a simpler form of exocytosis – we find that tethering is functionally coupled to fusion by the exocyst, such that fusion occurs rapidly and completely after tethering (half-time of ~7.5 s; Fig. 1h). Importantly, when we impair exocyst function by mutating Exo70 (i.e. Exo70-KK), we find that vesicles take twice as long to fuse (half-time of ~15 s; Fig. 1h) and often do so incompletely, by undergoing kiss-and-run instead of full fusion. An extreme case of exocyst impairment is when

vesicles are optogenetically tethered in the absence of the exocyst (i.e. via CRY2-Rab11 in Exo70 knockdown cells). In this case, we find that vesicles remain tethered for several minutes before kissing (Fig. 5). This scenario might loosely be thought of as a docked state, but it would be one that does not lead to full fusion.

The simplest explanation for these findings is that the exocyst promotes both tethering and fusion – in effect, coupling the two processes. This explanation is reasonable since 1) SNAREs alone may be insufficient to catalyze full fusion in living cells (Zick M & Wickner W. *Mol. Biol. Cell* 24: 3746–3753, 2013) and 2) the exocyst can promote SNARE complex assembly (Yue P, et al. *Nat. Commun.* 8: 14236, 2017). We speculate that additional mechanisms must be present in neuroendocrine cells to ensure that secretory vesicles can undergo full fusion, even long after they have docked. Thus, we would argue that studying constitutive exocytosis in HeLa is valuable, as it may help to clarify the mechanisms of exocytosis in regulated systems.

2. This reviewer cannot understand how the data in Figure 5 reveal that vesicle fusion contributes to membrane expansion. It might be helpful if the authors can provide evidence that active vesicle exocytosis is taking place at the site of membrane expansion.

For the reviewer's convenience, we reproduce Figure 6f below (left panel), in which we demonstrate that optogenetically triggered exocytosis promotes membrane expansion. Specifically, this figure shows that more fusion events occur during the two minutes right after stimulation (circles) than before it (crosses), particularly at the base of the expanding region (bottom left corner of the dashed box). **Based on the reviewer's suggestion, we have now plotted the time course of these fusion events. As depicted in the graph below (right panel), stimulation with 405-nm light (blue bar) induces a "burst" of exocytosis (red curve) that coincides with membrane expansion (black curve).**

It has long been proposed that exocytosis of recycling vesicles contributes to lamellipodial protrusion (Hopkins CR et al. *J. Cell Biol.* 125: 1265-1274, 1994; Bretscher MS & Aguado-Velasco C. *Curr. Biol.* 8: 721-724 (1998). Our optogenetics experiments support this long-held idea. However, we think it is unlikely that exocytosis contributes to lamellipodial protrusion simply by delivering membrane to the leading edge because the amount of membrane that vesicles deliver would not account for the observed increase in membrane area. Instead, we prefer the idea that exocytosis promotes membrane expansion through biochemical processes – for example, through interactions of the exocyst with actin regulators at the leading edge (Zuo X et al. *Nature Cell Biol.* 8: 1383–1388, 2006; Biondini M. et al. *J. Cell Sci.* 129: 3756–3769, 2016).

3. It is well recognized that the exocyst complex helps to tether and dock vesicles. However, this is not studied at all in this manuscript. In particular, the Exo70-KK mutant should significantly decrease vesicle docking. It seems that it would be important to examine docking in this study.

We agree that one might expect the Exo70-KK mutant to decrease vesicle tethering, given that Exo70-KK is deficient in membrane binding (He B et al. *EMBO J.* 26: 4053–4065, 2007) and inhibits exocytosis (Liu J et al. *Mol. Biol. Cell* 18: 4483–4492, 2007). However, we find that membrane binding by Exo70 is important for the fusion mode (Fig. 1g), not for tethering per se. In fact, to our surprise, Exo70-KK partially rescues the fusion rate in Exo70 knockdown cells, albeit by supporting kiss-and-run instead of full fusion (Fig. 1d). Nonetheless, our findings are consistent with the literature, as the reported inhibition of exocytosis by Exo70-KK (Liu J et al. *Mol. Biol. Cell* 18: 4483–4492, 2007) was based on an end-point assay of surface delivery of vesicle cargo, which of course depends on full fusion. We speculate that vesicles are still able to tether with Exo70-KK because Sec3 – which would be present in exocyst complexes containing Exo70-KK – can also bind membranes (Zhang X et al. *J. Cell Biol.* 180: 145–158, 2008). Indeed, a functional redundancy between Exo70 and Sec3 – one that is unique among the subunits – was described in fission yeast (Bendezú FO, Vincenzetti V & Martin SG, *PLoS ONE* 7: e40248, 2012).

As mentioned above, docking – as it is traditionally understood – likely does not occur in HeLa cells for two reasons: 1) exocytosis is not Ca^{2+} -dependent in HeLa cells (Jaiswal JK, Andrews NW & Simon SM, *J. Cell Biol.* 159: 625–635, 2002; Jaiswal JK, Rivera VM & Simon SM, *Cell* 137:1308–19, 2009) and 2) we observe fusion with newcomer vesicles. We note that the lifetime of all Sec8 spots at the membrane shows a median duration of only ~7.5 s in TIRFM experiments (Rivera-Molina F & Toomre D, *J Cell Biol.* 201: 673-680, 2013).

4. Why use 100 mM HEPES to determine the mode of fusion? In other studies, researchers use 5 versus 25 mM HEPES, so it is not clear why the authors of the current study use 25 and 100 mM HEPES.

To our knowledge, most researchers use extracellular buffers at 50–100 mM (HEPES or Tris) to determine fusion mode (Gandhi S P & Stevens C F, *Nature* 423: 607–613, 2003; Bowser DN & Khakh BS, *Proc. Natl Acad. Sci. USA* 104: 4212–4217, 2007; Zhang Q, Li Y & Tsien RW *Science* 323: 1448–1453 (2009), Zhang Z et al. *Mol. Biol. Cell.* 22: 2324–2336, 2011; Jullié D et al. *J. Neurosci.* 34: 11106-11118, 2014). The need for high buffer concentrations (≥ 50 mM) is demonstrated in a study of synaptopHluorin exocytosis in astrocytes (Bowser DN & Khakh BS, *Proc. Natl Acad. Sci. USA* 104: 4212–4217, 2007), which tested 2.5, 25, 60 and 100 mM HEPES. Here, the authors found that the difference in pHluorin decay time between 2.5 and 25 mM was nominal, compared to the difference in decay time between 2.5 and either 60 or 100 mM, with the latter showing the greatest difference.

The authors cite ref.37 for this assay. However, ref. 37 is a review from Richard Tsien; the original paper should be cited.

We now cite the original article demonstrating the use of extracellular buffers (Tris) to detect kiss-and-run (Gandhi S P & Stevens C F, *Nature* 423: 607–613, 2003) as well as the original article that demonstrated kiss-and-run with 100 mM HEPES (Bowser DN & Khakh BS, *Proc. Natl Acad. Sci. USA* 104: 4212–4217, 2007).

In addition, the pHluorin traces in full fusion events in this manuscript are very different from Ref. 37. The authors should explain this difference. Moreover, to my eye, the decay in Fig. 1F is faster in 100 mM HEPES, not slower. To be clear: on Page 8, the authors write, “100 mM HEPES affected TfRc-pH fluorescence only with Class ii events, ablating the faster decay of the receptor compared to ligand in 25 mM HEPES (dashed boxes in Fig. 1f)”. However, again, Fig. 1f shows that decay of the receptor is faster than ligand using 100 mM HEPES.

We are grateful to the reviewer for finding this error; the description of the result for Figure 1F in the text did not match what is shown as we mistakenly reversed the order of the 25 mM and 100 mM HEPES traces. We have now corrected this error.

Regarding the first point, the reviewer is correct that the pHluorin traces for full fusion events in our manuscript are different from Ref. 37 (Alabi AA & Tsien RW, *Annu. Rev. Physiol.* 75: 393–422, 2013). There are two important points about the full fusion event depicted in Ref. 37: 1) it is that of a synaptic vesicle and 2) it is detected by confocal microscopy. The fluorescence of synaptic vesicles in cultured neurons is punctate, reflecting the localization of vesicles in both terminal and *en passant* boutons. When a synaptic vesicle undergoes exocytosis and delivers a pHluorin-based reporter to the membrane, that reporter will remain, to a large degree, within the bouton – presumably due to geometric, compositional and functional considerations (e.g., synaptic vesicles recycle at protein-rich active zones within confined terminals). As such, after full fusion, the pHluorin signal may remain at the bouton for seconds and not disappear until its fluorescence is quenched by endocytosis of the reporter. This explains why the trace for a full fusion event of a synaptic vesicle can appear step-like.

On the other hand, the fluorescence of a vesicle in HeLa cells imaged by TIRFM is not as “compartmentalized.” In our experiments, when a vesicle fully fuses and delivers a pHluorin-based reporter to the membrane, the fluorescence of that reporter will laterally spread within the total membrane of the cell. As such, a full fusion event imaged by TIRFM will produce a trace that quickly decays after abruptly increasing.

5. The authors need to edit the paper more carefully. Quite a few sentences are difficult to understand. For example, on Page 7, the authors wrote, “With TfRc-pHTomato, the tethering duration was similar to that observed with Exo70-WT, Sec8 and Rab11, but when Exo70 KD was knocked down, the duration was anomalously long.” What is Exo70 KD?

Again, we thank the reviewer for carefully reading the manuscript. We have now corrected the text to read: “... but when Exo70 was knocked down, the duration was anomalously long.”

Overall, there are interesting results in the manuscript, but it is a bit difficult to understand some of the findings.

We thank the reviewer for the helpful comments and hope that we have clarified the manuscript.

REVIEWER COMMENTS

Reviewer #1 (Remarks to the Author):

In this resubmission, the authors have thoughtfully addressed most of the reviewers' concerns, and the addition of the iLID optogenetic system was very helpful in addressing several issues.

Reviewer #2 (Remarks to the Author):

The authors have made many improvements on the points I have mentioned. In particular, I recommended the author to apply other optogenetic systems because of the homo-oligomerization of CRY2, and it was impressive that activation of vesicle tethering is possible even with the iLID system. However, the next few points are still insufficient.

1. The iLID system of Fig. 3 does not show different results from the cry2-CIB system in Fig. 2. It seems like just another duplicated data in the main figure although it has few additional data. Therefore, I am not sure that it needs to be in the main figure.
2. I asked whether it is possible to control the fate of vesicle depending on various light conditions in regard to intensity and frequency of light. Instead, the author just answered that the study of exocyst can be applied to a wide range of studies.
3. Fig. 6 is insufficient in finding physiological meaning. The author appealed to another reviewer about the importance of this study, but it does not mean much except to support the long-held idea. There is no clear answer to the general question, and the reason for the appealing content is insufficient.
4. Also, the author had set the conditions for activation with 488 nm light in Fig. 6a-d, but it confuses me that the author used 405 nm light condition in Fig. 6e-f. As in Fig.6e-f, it is a great advantage to have different wavelengths of light for imaging and activating. It allows you to get the image before activating the cell with light. In particular, I wonder why the cell area decreases, as soon as activating the cell with light, so I want to check the image before light stimulation. Thus, I think it would be clear if the Fig.6 shows data with uniform conditions.

Reviewer #3 (Remarks to the Author):

In the original paper, the major problem is the use of HeLa cells for the study of exocytosis. In the revised manuscript, the authors did not address this major issue. The main conclusion from this manuscript is that the exocyst complex could promote not only vesicle tethering, but also membrane fusion and even expansion. Since this provocative conclusion could have a profound impact in this field, it would be important to demonstrate these findings in neuroendocrine cells or even neurons. HeLa cells are not so often used as a model system for exocytosis, thus raising the issue that the conclusion was due to the use of this particular cell line. As much less is known about exocytosis in HeLa cells, it is challenging to determine if the observed function of the exocyst complex is due to the use of new tools or simply a different cell line.

Moreover, it is unclear what types of vesicles are characterized in this manuscript. Also, it is critical to know how the findings in this study apply to exocytosis in general.

In conclusion, this study is developing new optical approaches to invoke an unknown function of the exocyst complex in not-so-well characterized secretion processes. Thus, it raised more questions that should be addressed. The results are interesting but need to be validated using a different approach in another type of cell.

Reviewer #1 (Remarks to the Author):

In this resubmission, the authors have thoughtfully addressed most of the reviewers' concerns, and the addition of the iLID optogenetic system was very helpful in addressing several issues.

We are grateful to the reviewer for the positive comments.

Reviewer #2 (Remarks to the Author):

The authors have made many improvements on the points I have mentioned. In particular, I recommended the author to apply other optogenetic systems because of the homo-oligomerization of CRY2, and it was impressive that activation of vesicle tethering is possible even with the iLID system. However, the next few points are still insufficient.

1. The iLID system of Fig. 3 does not show different results from the cry2-CIB system in Fig. 2. It seems like just another duplicated data in the main figure although it has few additional data. Therefore, I am not sure that it needs to be in the main figure.

We again thank the reviewer for requesting that we test the iLID system. While the iLID experiments represented a major undertaking, we feel they were worth the time and effort. This is because using a second, independent heterodimerization system greatly strengthens our main conclusion that membrane binding by Exo70 controls the fate of vesicles (i.e., mode of vesicle fusion) by ruling out a complication from potential CRY2 homo-oligomerization during optogenetically induced tethering with the CRY2-CIB system, which was the concern of the reviewer. In our opinion, the iLID experiments serve as an important extra degree of validation (which is why we agreed to perform them) and should be included in the manuscript as a main figure. However, we are happy to defer to the editor here.

2. I asked whether it is possible to control the fate of vesicle depending on various light conditions in regard to intensity and frequency of light. Instead, the author just answered that the study of exocyst can be applied to a wide range of studies.

We apologize if we misinterpreted the focus of the original question: ***“Are there any results that can be controlled by varying intensity or frequency conditions to control the fate of vesicles? If it is available, it could be applied to a wider range of studies.”*** We misinterpreted this to be if there were other cellular processes that could be controlled by controlling the fate of vesicles, which would broaden the applicability of our optogenetics system. We answered in the affirmative because the exocyst is implicated in numerous biological processes, such as cell migration, glucose transport, cytokinesis, ciliogenesis, autophagy and cell survival (Heider MR & Munson M. *Traffic* 13: 898–907, 2012).

But to the real nature of the question – we should have directly clarified that, yes, we have data showing that both intensity and frequency can control the fate of vesicles (we understand that the manuscript is rather dense and that this may have been overlooked). Namely, we showed the fate of vesicles can be controlled by modulating either the intensity of light (0.23, 1.5 and 2.6 W/cm²) in Fig. 4a or the frequency of stimulation (2 and 0.2 Hz) in Fig. 4b–d. In both cases, we found that suboptimal activation of tethering promotes kiss-and-stay (**KS**) rather than full fusion (**FF**). This idea – that the level of tethering controls the fate of vesicles – is supported by an independent approach later in the same

figure: when we reduce the number of exocyst complexes on vesicles by partially knocking down Sec8, we find that vesicles undergo kiss-and-run (KR) more often than FF (Fig. 4e, g).

We have now adjusted the writing to better emphasize that both light intensity and frequency can control the fusion mode and highlighted this in the discussion too.

3. Fig. 6 is insufficient in finding physiological meaning. The author appealed to another reviewer about the importance of this study, but it does not mean much except to support the long-held idea. There is no clear answer to the general question, and the reason for the appealing content is insufficient.

We agree with the reviewer that a long-held idea is that membrane protrusions such as lamellipodia arise from not only the force of actin polymerization but also the circulation of membrane derived from endosomes (Bretscher MS & Aguado-Velasco C, *Curr. Biol.* 8, 721–724, 1998; Hopkins C. et al. *J. Cell Biol.* 125, 1265–1274, 1994). However, we note that despite its proposal over two decades ago, this idea/concept has been challenging to test since it would require acute, rather than chronic manipulations and indeed others have discounted the role of membrane traffic here (Kay, RR et al. *Nature Rev. Mol. Cell Biol.* 9, 455–463, 2008).

Our data in Figure 6, which are based on the acute control of exocyst function using optogenetics, support that exocytosis of recycling vesicles – which are derived from the endosomal recycling compartment (Takahashi S. et al. *J. Cell Sci.* 125, 4049–4057, 2012; Rivera-Molina F & Toomre D. *J. Cell Biol.* 201, 673–680, 2013) – does promote membrane expansion, but only when there is full vesicle merger. These data dovetail with the known requirement for the exocyst in cell migration (Hertzog M, Chavrier P. *Biochem J.* 433, 403–409, 2011; Liu J & Guo W. *Protoplasma* 249, 587–597, 2012), which has been linked to cancer progression and metastasis (Camonis JH & White MA. *Trends Cell Biol.* 15, 327–332, 2005).

However, these are not the only reasons why Figure 6 has physiological meaning.

The key finding of our study is that vesicle tethering by the exocyst regulates the mode of vesicle fusion; without the exocyst, vesicles mainly undergo kiss-and-run (KR) instead of full fusion (FF). KR is an unconventional mode of fusion that has been the subject of investigation for decades in many cell types. The difficulty of detecting KR, let alone controlling it, has left many questions about this mode of fusion unanswered (for an excellent review on this topic, see Alabi AA & Tsien RW. *Annu. Rev. Physiol.* 75, 393–422, 2013). One fundamental question is whether there is a functional difference between KR and FF: that is, does it matter whether a vesicle undergoes KR or FF? Although the demonstrable difference in the release of cargo by these two fusion modes would suggest that there *must* be a functional difference, to our knowledge, a clear example of how a cellular process depends on the mode of fusion has yet to be demonstrated – again, because it has not been possible to control fusion mode up until now. As such, by showing that membrane expansion depends on optogenetic conditions that favor FF, we argue that Figure 6 provides a much-needed example of the physiological relevance of fusion modes.

Or posed another way – there is no data from others to date to suggest that the mode of fusion would impact membrane expansion.

We respectfully believe that showing an example of how the fusion mode may play a key cellular role helps to underscore the general importance of the mode of fusion – a hotly debated topic, especially in the neuroscience area (Alabi AA & Tsien RW. *Annu. Rev. Physiol.* 75, 393–422, 2013). We also recognize

that this is by no means any final answer to the question surrounding the role of exocytosis in membrane expansion – rather it lends support to a long-hypothesized concept and shows that fusion mode may be a key unrecognized regulator of this cellular process.

4. Also, the author had set the conditions for activation with 488 nm light in Fig. 6a-d, but it confuses me that the author used 405 nm light condition in Fig. 6e-f. As in Fig.6e-f, it is a great advantage to have different wavelengths of light for imaging and activating. It allows you to get the image before activating the cell with light. In particular, I wonder why the cell area decreases, as soon as activating the cell with light, so I want to check the image before light stimulation. Thus, I think it would be clear if the Fig.6 shows data with uniform conditions.

The reviewer raises a good point. The challenge here is that in order to use different wavelengths of light for imaging and activating cells, we would need to use an exocytosis reporter that can be excited with a wavelength longer than 488 nm, since 488 nm is the wavelength needed for activation. For most of our experiments though, it was necessary to image transferrin receptor-pHluorin (TfRc-pH), which is excited at 488 nm, to detect exocytosis. This is because it would not have been possible to reliably detect different fusion modes without pHluorin. Simply put, pHluorin (when localized within the vesicle lumen) is the best pH reporter of exocytosis as its pH sensitivity ($pK_a \sim 7.1$; Sankaranarayanan S. et al. *Biophys J.* 79, 2199–208, 2000) is unmatched when it comes to reporting the pH change that occurs within a vesicle during exocytosis (pH ~ 6.0 to 7.4).

However, in Figure 6, after determining that optogenetics conditions which favor FF cause membrane expansion (Fig. 6a–d), we wished to do our due diligence and image cells before activating them, to make sure that membrane expansion occurs only during the activation period. Since it was not necessary to detect fusion modes at this point, we imaged TfRc-pHTomato, which can be excited with 561-nm light. Thus, in Figure 6e–g, by imaging TfRc-pHTomato before and during CRY2 activation, we were able to show that membrane expansion occurs only when CRY2 is activated. We should note that 405 nm was used to activate cells in these experiments since TfRc-pHTomato is substantially excited at 488 but not 405 nm (Li Y & Tsien RW. *Nat. Neurosci.* 15, 1047–1053, 2012), and we wished to avoid any unnecessary photobleaching.

However, we agree, as the reviewer suggests, that it may be better to show data in Figure 6 with uniform conditions, to avoid any confusion regarding the wavelength used for activation. For this reason, we have moved Fig. 6e–g into the supplemental section as a separate figure (Supplementary Fig. 12).

Reviewer #3 (Remarks to the Author):

In the original paper, the major problem is the use of HeLa cells for the study of exocytosis. In the revised manuscript, the authors did not address this major issue. The main conclusion from this manuscript is that the exocyst complex could promote not only vesicle tethering, but also membrane fusion and even expansion. Since this provocative conclusion could have a profound impact in this field, it would be important to demonstrate these findings in neuroendocrine cells or even neurons. HeLa cells are not so often used as a model system for exocytosis, thus raising the issue that the conclusion was due to the use of this particular cell line. As much less is known about exocytosis in HeLa cells, it is challenging to

determine if the observed function of the exocyst complex is due to the use of new tools or simply a different cell line.

We agree that it would be interesting to expand these studies to neurons or neuroendocrine cells; however, we believe that this is outside the scope of this manuscript.

Moreover, it is unclear what types of vesicles are characterized in this manuscript. Also, it is critical to know how the findings in this study apply to exocytosis in general.

The exocyst is attached to vesicles via an interaction between the Sec15 subunit and the Rab11 GTPase (Zhang XM et al. *J Biol Chem.* 279, 43027–43034, 2004; Wu S et al. *Nat Struct Mol Biol.* 12, 879-885, 2005), which is anchored to the membrane in its GTP-bound state (Hutagalung AH & Novick PJ. *Physiol Rev.* 91, 119–149, 2011). Various data from our lab and elsewhere have shown that vesicles carrying exocyst subunits or Rab11 are derived from recycling endosomes (Langevin J et al. *Dev. Cell* 9, 365–376, 2005; Ward, ES et al. *Mol. Biol. Cell* 16, 2028–2038, 2005; Takahashi S. et al. *J. Cell Sci.* 125, 4049–4057, 2012; Rivera-Molina F & Toomre D. *J. Cell Biol.* 201, 673–680, 2013). Moreover, the transferrin receptor, which we also image, is well known to traffic through the endocytic recycling compartment near the Golgi (Yamashiro DJ et al. *Cell* 37, 789–800, 1984). Indeed, transferrin ligand is often used as a colocalization marker to assess whether a protein traffics through the recycling pathway (Mayle KM, Le AM, Kamei DT. *Biochim Biophys Acta.* 1820, 264–81, 2011).

To recap, we imaged vesicles labeled with five different markers of the recycling pathway: (i & ii) the exocyst subunits Exo70 and Sec8, (iii) Rab11, (iv) transferrin receptor and (v) transferrin ligand. Thus, we are confident that the vesicles characterized in our manuscript are those emanating from the endocytic recycling compartment.

In conclusion, this study is developing new optical approaches to invoke an unknown function of the exocyst complex in not-so-well characterized secretion processes. Thus, it raised more questions that should be addressed. The results are interesting but need to be validated using a different approach in another type of cell.

Again, we respectfully believe that expanding these studies to another type of cell is beyond the scope.

REVIEWERS' COMMENTS

Reviewer #2 (Remarks to the Author):

The authors have addressed most of the reviewers' concerns.